# Boundary layer structure characteristics under objective classification of persistent pollution weather types in the Beijing area

Zhaobin Sun[1], Xiujuan Zhao*[1], Ziming Li[2], Guiqian Tang[3], Shiguang Miao[1]

1. Institute of Urban Meteorology, China Meteorological Administration, Beijing 100089, China
2. Environmental Meteorology Forecast Center of Beijing-Tianjin-Hebei, Beijing 100089, China
3. State Key Laboratory of Atmospheric Boundary Layer Physics and Atmospheric Chemistry, Institute of Atmospheric Physics, Chinese Academy of Sciences, Beijing 102300, China

*Correspondence to:* Xiujuan Zhao(xjzhao@ium.cn)

**Abstract.** Different types of pollution boundary layer structures form via the coupling of different synoptic systems and local mesoscale circulation in the boundary layer; this coupling contributes toward the formation and continuation of haze pollution. In this study, we objectively classify the 32 heavy haze pollution events using integrated meteorological and environmental data and ERA-Interim analysis data based on the rotated empirical orthogonal function method. The thermodynamic and dynamic structures of the boundary layer for different pollution weather types are synthesized, and the corresponding three-dimensional boundary layer conceptual models for haze pollution are constructed. The results show that four weather types mainly influence haze pollution events in the Beijing area: (a) type1: southerly transport, (b) type2: easterly convergence, (c) type3: sinking compression, and (d) type4: local accumulation. The explained variance in the four pollution weather types are 43.69%(type1), 33.68% (type2), 16.51%(type3), and 3.92% (type4). In persistent haze pollution events, type1 and type2 surpass 80% on the first and second days, while the other types are present alternately in later stages. The atmospheric structures of type1, type2, and type3 have typical baroclinic characteristics at mid-high latitudes, indicating that the accumulation and transport of pollutants in the boundary layer is affected by coupled structures in synoptic-scale systems and local circulation. The atmospheric structure of type4 has typical barotropic characteristics, indicating that the accumulation and transport of pollutants is primarily affected by local circulation. In type1, southerly winds with a specific thickness and intensity prevail in the boundary layer, which is favorable for the accumulation of pollutants in plain areas along the Yan and Taihang Mountains, whereas haze pollution levels in other areas are relatively low. Due to the interaction between weak easterly winds and the western mountains, pollutants accumulate mainly in the plain areas along the Taihang Mountains in type2. The atmospheric vertical structure is not conducive to upward pollutant diffusion. In type3, the heights of the inversion and boundary layers are the lowest due to a weak sinking motion while relative humidity is the highest among the four types. The atmosphere has a small capacity for pollutant dispersion and is

favorable to particulate matter hygroscopic growth; as a result, the type3 has highest $PM_{2.5}$ concentration. In type4, the boundary layer is the highest among the four types, the relative humidity is the lowest, and the $PM_{2.5}$ concentration is relatively lower under the influence of local mountain–plain winds. The findings of this study allow us to understand the inherent difference among heavy pollution boundary layers; in addition, they reveal the formation mechanism of haze pollution from an integrated synoptic scale and boundary layer structure perspective. We also provide scientific support for the scientific reduction of emissions and air quality prediction in the Beijing-Tianjin-Hebei region of China.

## 1 Introduction

Over the past 40 years, rapid industrialization and urbanization have caused serious haze pollution problems in China. Pollutants not only affect the climate system but also reduce visibility, affect city operation, and have a significant negative impact on human health. Haze pollution creates health costs for residents (Dockery et al., 1993; McDonnell et al., 2000) and emissions reductions costs (D'Elia et al., 2009).Governments must play a more flexible role and adopt an optimized strategy between health costs and emissions costs based on national or local economic affordability to reduce emissions (Lee et al., 2016).From an operability perspective, the timings of different emissions reductions strategies are largely dependent on trends in atmospheric pollution dispersion conditions (Zhai et al., 2016). Haze pollution is the combined effect that excessive emissions and adverse meteorological conditions have on the dispersion of pollutants (He et al., 2013; Li et al., 2017). With relatively few changes in the emission source, the diffusion conditions largely determine the duration and pollution level of a haze event.

First, from an atmospheric circulation perspective, persistent haze pollution generally corresponds to persistent adverse meteorological conditions for pollutant dispersion (Zheng et al., 2015), where persistent anomalies in atmospheric circulation are an important background (Inness et al., 2015).These conditions cause stabilized vertical stratification and low horizontal wind speeds (Chamorro et al., 2010; Park et al., 2014), such that the combination of these two conditions form "calm weather." From a large-scale climate circulation perspective (Markakis et al., 2017; Zou et al., 2017),previous studies have suggested that, if global warming trends continue, the probability of adverse atmospheric pollutant dispersion will continue to increase (Cai et al.,2017), where the reduction in sea ice can lead to the weakening of the rossby wave activity south of 40°N, rendering the lower layer colder and a reduced moisture content, a stable atmosphere, weaker wind speeds, and an increased chance of heavy haze pollution(Wang et al.,2015;Chen et al.,2015).These results show that the troposphere in the Beijing-Tianjin-Hebei area can produce a continuous deep downdraft under flat circulation or a weak high-pressure system, along with the boundary layer's southerly wind yielding the temperature inversion height and decrease in the atmospheric capacity, which provides a favorable dynamic condition for the maintenance and aggravation of haze pollution (Wu et al., 2017). Zhang et al. (2016) use the Kirchhofer technique to classify the circulation patterns, examining the influence that the monsoon has on the occurrence frequency of different weather patterns and air quality.

Second, the pollutant concentration also depends on local mesoscale circulation coupled with a stable boundary layer and synoptic-scale system (Miao et al., 2017),for example, valley wind, sea–land wind, heat island circulation, and mountain–plain wind. Even under conditions associated with weaker synoptic scales, these mesoscale systems largely determine the peak concentration and spatial-temporal distribution of the pollutants (Miao et al., 2017; Li et al., 2019). Previous studies have examined the interaction between aerosols and the boundary layer (Wu et al., 2019; Zhong et al., 2018; Wang et al., 2018a; Wang et al., 2018b; Zhou et al., 2018).Ding et al. (2016) find that black carbon aerosols play a key role in reducing the height of the boundary layer and enhancing haze pollution. Huang et al. (2018) investigate the interaction between aerosols and the boundary layer in North China using long-term observational data, quantifying the contribution of aerosols to the heating of the top layer of the boundary layer and cooling of the surface layer. Millan et al.(1997), studied the mechanism of aerosol transport back and forth along the coast under the combined action of weather system, sea-land winds and slope wind. In coastal cities of West Africa， Adrien et al. (2019) simulated the transport and mixing processes of biomass combustion aerosols in the boundary layer and at the top of the boundary layer under the action of dry convection a nd sea breeze front. Tobias et al.(2017) studied pollution in coastal valley cities (Bergen, Norway) , where the concentration of pollutants is determined by both large-scale topography and small-scale sea-land winds, when there is a strong background wind, the sea-land wind will submerge in the large-scale circulation, and the large-scale circulation and the local circulation in the boundary layer will cancel each other, causing ground-level air to stagnate and pollution levels to rise.

In summary, previous studies have achieved results in the study of the influence that the weather system and boundary layer have on the concentration of aerosols. A comprehensive analysis of the these two aspects, that is, combining weather systems and the structure of the boundary layer, however, is still rare. Liao et al. (2018) use the Self-Organizing Map method to classify the boundary layer in the Beijing area, as well as to examine the relationship between the classification results and pollutant concentrations. Miao et al. (2017) and Xu et al. (2019) use the obliquely rotated principal component analysis in T-mode (T-PCA) approach to classify synoptic patterns, analyzing the structure of the boundary layer and concentration of surface pollutants under different weather types in summer. The Beijing area is located in the transition zone between the plain and mountainous areas, with mountains to the west, north, and east. The southeastern region of Beijing is a flat plain that slopes toward the Bohai Sea. More than 20 million people live in Beijing who are affected by both the weather system and local circulation in the boundary layer. To formulate optimized emissions reduction strategies, we must master the main control factors that affect the haze pollution diffusion conditions in Beijing under different weather and boundary layer conditions. At present, under the influence of different haze pollution weather types, there are still a lack of studies on the three-dimensional haze pollution structure of the boundary layer, especially as the structure of the heavy haze pollution boundary layer is not entirely identical. The above-mentioned weather classification method does not take into account the continuity of the one haze pollution event, such as the first day of pollution weather pattern is same as the second day? what is the difference in the structure of the boundary layer between haze pollution weather types? The different structures of the boundary layer correspond to the different accumulation characteristics and pollutant efficiencies. However, previous studies did not unravel  structure differences of the heavy haze pollution boundary layer. Based on the objective classification of the

pollution weather types, we examine the boundary layer structures of different pollution synoptic types, revealing that the thermal and dynamic mechanisms of the boundary layer structures inhibit the diffusion of atmospheric pollutants. Based on the two interrelated dimensions, that is, the weather system and boundary layer structure, we systematically investigate the meteorological mechanism of haze pollution formation.

## 2Data and methods

### 2.1 Meteorological data

The weather classification data were derived from the ERA-Interim data from 2014–2017. ERA-Interim (0.125°×0.125°) is a new reanalysis data from the ECMWF (European Centre for Medium-Range Weather Forecasts ) after the ERA40, with 60 vertical layers and partially overlaps with the ERA40 in time. However, significant progress has been made in data processing, for example, from the three-dimensional assimilation system(3-D VAR) to the four-dimensional assimilation system(4-D VAR).The model parameters were changed, and the horizontal resolution was enhanced with the use of more satellite and ground-based observations(https://apps.ecmwf.int/datasets/).

The 850hPa geopotential height field (30–50°N, 110–128°E) of the ERA-Interim was used to classify the weather system. The meteorological elements at 850hPa interact with the meteorological elements in the boundary layer. At the same time, the 850hPa is evidently influenced by the free atmosphere, especially in Beijing area, which can be regarded as the transition layer between local thermal circulation (valley wind, sea–land wind, and mountain–plain wind ) and the free atmosphere. In addition, the hourly relative humidity, visibility, and wind speed observed at the Beijing Observatory (39.93°N, 116.28°E) were used in this study.

A 12-channel (5water channels and 7oxygen channels) microwave radiometer (Radiometrics, Romeoville, IL, U.S.A.) was used to measure the relative humidity and temperature profile in the atmosphere. The microwave radiometer was installed in the Beijing Observatory (39.93°N, 116.28°E) and was calibrated every three months. The wind profiles, including the wind speed and direction between 100 and 5,000m, are measured at the same station by a wind profiler. The wind profiler radar provides a set of profile data every 6min at a detection height of ~12–16km.

### 2.2 Air quality monitoring data and haze pollution event definition

Hourly $PM_{2.5}$ concentrations at 12 national stations and the daily air quality index (AQI)in Beijing are available from http://zx.bjmemc.com.cn/?timestamp=1564483254009. Surface $PM_{2.5}$ mass concentrations were measured by the tapered element oscillating microbalance method. The measurements were calibrated and quality controlled according to the Chinese environmental protection standard (HJ 618-2011).

As this study focuses on episodes of heavy haze pollution, we first defined the criteria. Haze is defined by the relative humidity and visibility; therefore, the haze pollution level is defined by the AQI and the primary pollutant. Considering that

haze pollution mainly refers to reduced visibility caused by fine particulate matter, as well as taking into account the effects of the pollution levels and duration, the screening criteria for heavy haze pollution were still based on the AQI, $PM_{2.5}$ concentration, and the duration of low visibility. The specific criteria of a haze pollution event can be defined as follows: the AQI reaches a moderate pollution level (AQI≥150) for more than or equal to 3 days and at least 1 day reaches the heavy pollution level (AQI>200). The primary pollutant is $PM_{2.5}$ in Beijing area. As defined by the AQI, the 24-h average concentration of $PM_{2.5}$must be above 115 μg $m^{-3}$ for more than three consecutive days and above 150 μg $m^{-3}$ for at least 1 day. At the same time, the accumulated time of horizontal visibility, that is, less than 5 km, has a duration of at least 12 h each day at the Beijing Observatory station.

Based on these criteria, 32 events (125 days) were screened for heavy haze pollution in Beijing between 2014 and2016. Eight events occurred in spring and summer while 24 events were concentrated in autumn and winter, 32 events accounting for 75% of the events that occurred during the study period (2014–2016). We collected ground-based routine meteorological observation data in North China, L-band radar second-order sounding data (including wind, temperature, and humidity), wind profile data, ceilometer data, and tower data during these events.

### 2.3 Attenuated backscattering coefficient measurements and boundary layer height calculation

#### 2.3.1 Attenuated backscattering coefficient measurements

We used the CL31 and CL51 Vaisala-enhanced single-lens ceilometer instrument, which uses the pulse diode laser LIDAR (laser detection and ranging) technology to measure the backscattering profile of atmospheric particles and the cloud height. The main parameters of the CL31 and CL51 are respectively as follows: range of 7.6 and 13 km, reporting periods of 2–120 and 6–120s, reporting accuracy of 5and 10m/33ft, peak power of 310w, and wavelength of 910mm. The geographic location of the station is 39.974 °N and 116.372 °E, with an elevation of approximately 60 m (Tang et al.,2016).

#### 2.3.2 Boundary layer height calculation

As the lifetime of a particles is long, that is, several days or weeks, the particle concentration distribution in the boundary layer is more uniform than that of the gaseous pollutants, whereas the particle concentration in the boundary layer is significantly different from that in the free atmosphere. By analyzing the backscattering profile of the atmospheric particles, we located the abrupt change in backscattering at the top of the boundary layer.

This study used the gradient method (Christoph et al., 2007; Zhang et al., 2013; Tang et al., 2015) to determine the boundary layer heights. The maximum negative gradient in the aerosol backscattering coefficient profile occurs at the top of the boundary layer, but is easily disturbed by data noise and the aerosol structure. Therefore, we must select a continuous region of time or space for averaging to smooth the contour map vertically after averaging and adopt an improved gradient (http://isars2010.uvsq.fr/images/stories/posterexabstracts/p_bls06_muenkel.pdf) method to manage severe weather (such as

precipitation and fog). Despite this, the gradient method still has certain defects, especially for neutral atmospheric stratification, where the inverse calculation of the boundary layer height is not accurate.

### 2.4 Objective classification of pollution weather types

Using the ERA-Interim reanalysis data, the 925hPa geopotential heights of all pollution events in this study were analyzed with the rotated empirical orthogonal function (REOF) to determine which mode the pollution events belong to according to the characteristic values of the different pollution events for determining the days characterized by specific types of pollution weather. Since Lorenz(1956) introduced empirical orthogonal function (EOF) analysis to atmospheric science, this simple and effective method has been widely used in atmospheric, oceanic and climatic studies. The essence of EOF analysis is to identify and extract the spatiotemporal modes that are ordered in terms of their representations of data variance（Lian et al.,2012）. In the empirical orthogonal function (EOF) analysis, the first few main components are the focus of the analysis element variance, such that the EOF method can highlight the entire correlation structure of the analysis element. However, the local correlation structure is not sufficient, which is a defect of the pollution weather classification based on the EOF. The spatial patterns (EOFs) and the temporal coefficients of these modes are orthogonal. This orthogonality has the advantage of separating unrelated patterns, but it sometimes leads to the complexity of spatial structure and the difficulty of physical interpretation (Hannachi, 2007). Based on the EOF analysis, the REOF transforms the load characteristic vector field into a maximum rotation variance, as a result of which each point in the rotation space vector field is only highly correlated with one or a few rotation time coefficients. Previous studies have shown that REOF analysis can avoid non-physical dipolelike EOF analysis patterns, which often occur when known dominant patterns have the same symbols in the region (Dommenget et al.,2002). REOF analysis outperforms EOF analysis almost certainly in reconstructing spatially overlapped modes, and that this superiority is not sensitive to parameters such as the number of modes, the spatial scale of the signal, and the degree of rotation（Lian et al.,2012). Thus, the high load value areas are concentrated in smaller areas, while the remaining areas are relatively small and nearly 0, highlighting the pattern and characteristics of the abnormal distribution of elements (Paegle et al., 2002; Chen et al., 2003), the classification of heavy pollution weather types based on this method is more consistent with the requirements of this study. Pollution weather types were classified by the REOF method to analyze the differences in the structures of the pollution boundary layer.

## 3Results and discussion

### 3.1 Pollution weather type classification and horizontal characteristic analysis

In this study, the 925hPa geopotential height was used to classify the pollution weather types into four categories with the REOF method, as shown in Fig. 1: (a) type1, that is, influenced by southerly winds at the rear of the high pressure system, (b) type2, that is, influenced by easterly winds at the bottom of the high pressure system, (c) type3, that is, a weak downdraft

effect in the high pressure system, and (d) type4: no significant weather system. In this study, we observed 125 days of heavy polluted weather. Among these days, type1, type2, type3, and type4 had 67, 27, 21, and 10 days, respectively (Fig.2), where the four weather types accounted for 53.6, 21.6, 16.8, and 8.0% of the total sampled weather event days, respectively. The total interpretation variance of the four types for all events was 97.8% while the independent interpretation variance was 43.69, 33.68, 16.51, and 3.92%, respectively (Fig. 2). This indicates that an objective weather classification can effectively obtain the main feature information of the pollution weather types.

As shown in Fig. 1, the Beijing area is located toward west of the high-pressure system that has its center located in the sea. The low pressure system is located in the northern Hebei province for type1, where southerly winds control the 925hPa, which is favorable for the regional transportation of pollutants. When type2 appears, the Beijing area is located at the bottom of the high pressure system in Northeast or North China. In the plain area, the sea level pressure in the eastern part of Beijing is higher than that in the central Beijing area, such that there is an evident pressure gradient. Due to pressure-gradient forcing, the boundary layer appears within the easterly wind component while the easterly wind speed is smaller, which leads to pollutant convergence into the plains along the Taihang Mountains, When type3 appears, the high pressure center was located in the middle of Mongolia, where Beijing was in the front of the weak high pressure system, with a northwest current at 925hPa (Fig. 1i). However, the wind speed was lower than that affected by strong cold air, because of which it was difficult to penetrate the lower layer of the boundary layer and the wind can only exist in the upper atmosphere of the boundary layer. When type4 appears, the center of the high pressure system is located further to the north in the western part of Mongolia and southern Hebei province, where there is only a low pressure system with a smaller spatial and temporal scale. On the other hand, the synoptic-scale low pressure system is already located over the sea in the eastern Jianghuai region, showing that the high and low pressures corresponding to the synoptic-scale system are far from the Beijing area, which results in a smaller synoptic-scale pressure gradient in Beijing and the surrounding areas (Fig. 1i). Most areas in North China do not have strong weather systems and the average wind speed of the boundary layer is smaller, which is favorable to the formation and maintenance of the local circulation considering the topography in the Beijing area. The wind speed of type4 is more difficult to determine via the evolution of the wind field in the lower boundary layer based on the effect of descending momentum. Therefore, the dynamic pollutant process in the boundary layer in type4 is more related to the local circulation.

### 3.2 Vertical thermal and dynamic structure characteristics under four weather types

The vertical structure of the atmosphere is very important for the formation and evolution of extreme haze events. The vertical thermal and dynamic structures of four weather types are investigated in three-dimensional view. Figure 3 to Figure 6 presented the vertical distribution of temperature, wind and RH, respectively. To classify the pollutant regimes according to the various meteorological features, we summarized relevant thermodynamic and dynamic parameters in Table S1.

Figure 3 shows that the strong inversion is located at 800–900hPa for type1. In type2, easterly winds with low temperatures influence the temperatures below 800hPa, where a cooling layer appears at 900 hPa, with the height of inversion between 700 and 800hPa. The inversion height for type3 is the lowest among the four types due to the sinking motion, where the inversion is mainly below 900hPa, which causes a rapid decline in the atmospheric capacity. The atmospheric structure is also relatively stable in type4, whose inversion structure is similar to type2.The mechanism of the thermal structure, however, is different, where the inversion height of this type is between 700 and 800hPa.

As shown in Fig. 4, the basic flow is the southerly wind below 2,000m in type1, where a southwest wind appears from 500–2,000m. The southerly wind is below 500m between 04:00 and 20:00, and the easterly wind appears at other times. The southerly wind speed at 500m is strong, while the easterly wind is weak. In type2, the basic flow above 1,000m is westerly wind, where the layer between 500 and 1,000m is a weak wind layer. We note that the wind velocity in this layer is the smallest when there is an increase in the easterly component below 500m.This indicates that the weak wind layer is the wind shear transition layer between the westerly wind above 500m and the easterly wind below 500m. The easterly and westerly winds cancel each other at this height and form a small wind velocity layer. From 04:00 to 20:00, southerly winds appear below 500m while we observe the appearance of easterly winds at other times. The space-time structure of the wind field below 500 m was similar to that of type1, but the southerly wind speed was lower than in case of type1.In type3, the wind above 500m originates from the northwest from 04:00 to 14:00. At altitudes below 500 m, the wind is southerly and northerly at other times. Whether it is southerly or northerly, the wind speed is smaller. Mountain–plain wind in the Beijing area causes this diurnal and nocturnal circulation of the wind field. In type3, the wind velocity below 500m is less than that of type1 and type2, because the basic flow is northerly, where northerly wind superposes onto the plain wind (southerly), which may weaken the southerly wind speed. The observed data are the superposition results of two scale wind fields (i.e. local circulation and basic airflow). Westerly or weak northerly winds above 1,000m in type4 control the atmosphere, where the wind velocity below 1,000m is significantly weak. For the majority of the time, the wind velocity is less than 4 m $s^{-1}$, but the mountain–plain diurnal cycle wind can still be observed from the diurnal variation in the wind direction. From 08:00 to 18:00, the wind is southerly while mainly northerly at night. Weak wind speeds last for a long period in the boundary layer of type4, because of which the local thermal and dynamic conditions can become the main factors that affect the spatial-temporal distribution of haze pollutants in Beijing.

Figure 5 shows that, below 700hPa, type1, type2, and type4 are ascending movements. The maximum of the synoptic scale ascending movement appears in 900–950hPa.With an increase in the height, the intensity of the ascending movement gradually weakens, whereas in type3, below 750hPa can be characterized as a sinking movement. The intensity of the sinking movement increases gradually with decreasing height, where the maximum of the sinking movement appears at 900–950hPa.The intensity of the subsidence movement from this layer at 900–950hPa to the ground decreases a second time. Therefore, the sinking movement affects the inversion layer of type3, where the height of the inversion layer is the lowest of all types, resulting in type3 characterized by the smallest capacity among the four types.

Based on Figure 6, the relative humidity profiles for the four weather types have both similarities and differences in their space-time structures. The similarities in the four types are the increased and decreased relative humidity below 1,000m during the night and day, respectively, with a reverse in the relative humidity layer appearing during the day. The relative humidity of the surface layer decreases daily from 10:00 to 20:00 with an increase in the solar radiation. The thickness of the dry layer in the surface layer increases continuously, reaching its maximum height at ca. 14:00 or 15:00 every day, but the maximum height of the dry layer does not exceed 500m. The top of the dry layer is the reverse of the relative humidity layer. Above 1,000m, the relative humidity of the other three types, except type2, decreases significantly during the day.

The difference in the relative humidity field among the four types can be summarized as follows. The average relative humidity below 1,000m is higher than that above 1,000m. The inverse relative humidity structure appears below 500m in type2 and type3 from 00:00 to 05:00, with a maximum relative humidity center of more than 90%. Above 500m, the relative humidity also increases from 05:00 to 12:00. The relative humidity structures of type1, type2, and type3 all contain a baroclinic structure from lower to higher levels, where the baroclinic structure in type2 is more evident because the basic flow in type2 is westerly, which reflects the baroclinic characteristics of the atmosphere in the mid-high latitudes of East Asia. The basic flow is generally westerly in this area, where type1 and type3 are more typical of the disturbances in the northerly and southerly wind in the westerlies, which is the fluctuation feature of the basic flow. The relative humidity profile in the pollution boundary layer formed under the condition of wave-current interaction in the atmosphere (Fig. 6).

Type2 has strong westerly characteristics (Fig. 4), which reflects more baroclinic characteristics in the atmospheric vertical structure for the westerlies. Based on the analysis of the wind field, type4 is characterized by an average wind speed that is the weakest among the four types. Three important factors determine the baroclinicity, that is, the density gradient, pressure gradient, intersection angle between the density surface and pressure surface. This may be an important factor why relative humidity field has more barotropic characteristics. From the analysis of the baroclinic and barotropic characteristics, we can observe that the weather systems of type1, type2, and type 3 have a significant influence on the accumulation and transport of pollutants in the Beijing area. The mountain–plain wind in type4 can occur due to weakening in the weather system (Fig. 4).

### 3.3Construction of 3-D conceptual model for the pollution boundary layer

Based on the characteristics of the circulation field and the vertical thermodynamic structure for the four weather types, we established conceptual models of the boundary layer structure under the influence of the four pollution weather types is established, which are: (a) type1: southerly transport; (b) type2: easterly convergence; (c) type3: sinking compression; (d) type4: local accumulation (Fig. 4). When type 1 appears, the Beijing area is located at the rear of the high-pressure system, consistent with southerly winds throughout the atmosphere, and multilayer inversion occurs in the boundary layer. Under the influence of a southerly wind, haze pollutants accumulate in front of the Yan and Taihang Mountains. The air pollutants in the Hebei region have evident regional transport features. When type2 appears, the Beijing area is located at the bottom of the high-pressure system, where the air above 850hPa is a westerly wind, with easterly winds below 850hPa. Under the

influence of easterly winds below 850hPa, haze pollutants tend to accumulate in front of the Taihang Mountains. The cross-mountain air mass flows from west to east, preventing the further dispersion of air pollutants in front of the Taihang Mountains. When type3 appears, a weak high-pressure system controls the Beijing area. A weak subsidence northwest flow influences the atmosphere above 850hPa, which further compresses the capacity of the atmosphere to absorb pollutants in the boundary layer. The southerly wind at 850hPa is favorable for pollutant transportation in the region and accumulation in front of the Yan and Taihang Mountains. The atmospheric vertical structure in the high-level northwest wind and low-level southward wind provides excellent conditions for the stability of atmospheric stratification with respect to dynamic conditions and a thermal structure. The 850hPa southerly winds favor regional pollutant transport and their accumulation in the area along the Yan and Taihang Mountains. The atmospheric vertical structure of the high-level northwest wind and low-level southerly wind provides excellent conditions for stratification stability in terms of dynamic-thermal structures because southerly wind at 850hPa is warm advection, where advection inversion can form in the boundary layer, while weak subsidence above 850hPa can cause subsidence inversion. These two inversion mechanisms are coupled at the interface between the northwest wind and southerly wind, resulting in stable atmospheric stratification. When type4 appears, there is often no evident synoptic-scale system surrounding Beijing, with a weak pressure gradient above 850hPa. Therefore, the average wind speed is weak. The most important local circulation in Beijing, that is, the mountain–plain wind, begins to form in the boundary layer and plays an important role in the spatial and temporal distribution of atmospheric pollutants, with the wind direction continuously shifting from the south to the north. The air pollutants accumulate near the terrain convergence line formed by the mountain–plain wind. The terrain convergence line also swings from north to south, such that air pollution in the Beijing area often appears as a “different sky” relative to a clean sky in the north and a polluted sky in the south.

## 3.4Effects of the four pollution weather types

### 3.4.1 Statistical analysis: effects of the four weather types on haze pollution

Figure 8 shows the statistical characteristics of the $PM_{2.5}$ concentrations and meteorological elements in terms of the four polluted weather types. The daily average $PM_{2.5}$ concentration in type3 is the highest at 245 μg $m^{-3}$ and type4 is the lowest at 181μg $m^{-3}$ (Fig. 4). The daily average relative humidity values of the four pollution weather types are >60%, with a maximum relative humidity of 72.3% in type3 and a minimum relative humidity of 63.5% in type4 (Fig. 8b).Under the influence of a high relative humidity and high $PM_{2.5}$ concentration, the daily average visibility for the four heavy pollution weather types is less than 4,000m, with a minimum daily average visibility of 2,193m in type1. The maximum daily average visibility is 3,624m in type4 (Fig. 8c). The mean 24h wind speeds for the four pollution weather types are all less than 2.0 m $s^{-1}$.

The mean daily wind speeds of type1 and type3 are both smaller, that is, 1.38 and 1.49 m $s^{-1}$, respectively. The mean daily wind speeds of type2 and type4 are relatively faster, that is, 1.70 and 1.76 m $s^{-1}$, respectively (Fig. 8d). There is a significant

negative correlation between the boundary layer height and $PM_{2.5}$ concentration. The lowest boundary layer height was 386.5 m for type3, followed by type1, whereas type4 had the highest boundary layer height.

In this study, we calculated the distribution of the weather types from the first to last day of the persistent haze pollution events (Fig. 9). The daily synoptic types from the first to eighth day of persistent haze pollution events were calculated. As the number of pollution events that lasted more than five days is relatively small, the classification results were combined with the statistics for the events defined as greater than or equal to five days. The results show that the cumulative proportion of type1 and type2 occurrences on the first and second pollution day are more than 80%, indicating that regional transport plays a more prominent role in the initial stage of haze pollution formation, which is consistent with previous analyses (Zhong et al.,2018).On the third day and thereafter, the proportion of type1 began to decrease, but still exceeded 30%. Type2, type3, and type4 began to alternately affect the Beijing area. This indicates that, after the first and second days, the center of high pressure over East China in type1 began to move eastward away from the mainland. Beijing is located at the rear of the high-pressure system, where the $PM_{2.5}$ concentration corresponding to type1 increases throughout most of the day. The timing of the initial rise in the $PM_{2.5}$ concentration is the earliest among the four types, which indicates the role of the rear within the high-pressure system in the transmission of pollutants (Fig. 10a). When the upstream weather system begins to affect the Beijing area, it is occasionally located at the bottom of the high-pressure system (type2). The diurnal variation in the $PM_{2.5}$ concentration in type2 was similar to the mean annual variation in the $PM_{2.5}$ concentration in the Beijing area. The first peak was at 10:00 and the second was at 20:00(Zhao et al., 2009) (Fig. 10a). The weak high-pressure system in type3 can directly affect the haze pollution diffusion conditions in the Beijing area, but the intensity of the cold air behind the upper trough is weak. The $PM_{2.5}$ concentration in type3 is higher at night and lower during the day, with the highest average $PM_{2.5}$ concentration among the four types. Based on this analysis, we can observe that, in type3, the height of the inversion layer is the lowest and the atmospheric capacity to contain pollutants is also the lowest under the influence of a weak downdraft (Fig. 10a). In type4, there is no evident weather system that affects the Beijing area. An increase in the thermal difference between the mountain and plain affects local circulation development. The average $PM_{2.5}$ concentration in type4 is the lowest among the four types. The diurnal variation in the $PM_{2.5}$ concentration shows a typical "v" pattern. After sunrise, the $PM_{2.5}$ concentration begins to decrease while, after sunset, the $PM_{2.5}$ concentration increases significantly, which was due to the fluctuation of aerosols under local meteorological conditions (Fig. 10a). Based on Fig. 10b, the boundary layer height of type3 is the lowest among all types for most part of a day, which is mainly related to the suppression of the weak synoptic-scale downdraft. The change in the trend of the boundary layer height is similar to that type2 and type4 for most of the day. However, the boundary layer height is less developed when the thermal conditions are strongest between 12:00 and 18:00, which is similar to type3. The boundary layer heights of type2 and type4 are relatively high, and the corresponding $PM_{2.5}$ concentrations are the lowest out of the four pollution types (Fig. 10b).

The above analysis shows that in one persistent multi-day pollution event, the weather patterns that affect the Beijing area change daily, that is, they also change according to the basic principles of synoptic dynamics, which is the natural development and evolution of rossby waves in the mid-high latitude westerly belt. This also indicates that it is not

appropriate to classify a multi-day pollution event as a defined type (such as the low-pressure or high-pressure type). We cannot rule out the possibility that a pollution event may occur for several consecutive days under the influence of a low-pressure system, which is a rare event. Even then, this may also be a combination of different low-pressure systems. In addition, we note that, in one persistent multi-day heavy pollution event, different types of pollution weather types are linked together in a permutation that affects the structure of the boundary layer and thus the change in the $PM_{2.5}$ concentration (Fig. 9). As different types of weather systems form haze pollution events, we discuss the type of boundary layer structure formed by certain weather systems in the Beijing area and how this boundary layer structure influences the evolution of haze pollution formation.

**3.4.2 Effects of four weather types on the 3-D spatial-temporal evolution of haze pollution**

Figure 11 shows the aerosol vertical distribution under the influence of the boundary layer structure for the four pollution weather types. The wind below 2,000m for type1 in Figure 11 is southerly (Fig. 4), which facilitates regional pollutant transport. From 10:00 to 11:00, a v-shaped notch appears in the vertical structure of the aerosol at a height of 500–1,000m, which shows that there is a decrease in the extinction ability of the entire atmosphere below 1,000m. The boundary layer height rises above 1,000m from 11:00 to 17:00, showing an improvement in the local haze pollutant dispersion condition in Beijing, but the aerosol below 1,000m increases, which is more evident below 500m. This indicates that extrinsic aerosols are transported to Beijing area, which is consistent with the transport characteristics of southerly wind in the entire type1 atmosphere. Under the influence of southerly winds, the sensitive source areas related to the Beijing area are generally the plain areas along the Taihang Mountains in Hebei province (Wang et al., 2017). According to the dynamics, the positive vorticity advection in the direction of Beijing forms in the plain area. The positive vorticity advection in this boundary layer has two functions. First, the positive vorticity airflow is affected by the friction, coriolis effect, and pressure-gradient force. Second, the positive vorticity advection continuously transports the converging space field to the Beijing area and, at the same time, also transfers a large amount of external pollutants. The above analysis can explain the significant increase in the $PM_{2.5}$ concentration in the surface layer and the corresponding increase in the number of aerosols within 1,000m, which is a common phenomenon during regional haze pollution events in Beijing, Hebei, and Tianjin. However, westerly or weak northwest winds occur above 1,000m in type2. The dynamic stratification structure between the upper and lower layers is not favorable for downward momentum transfer, which results in the strengthening of southwesterly winds in the boundary layer (Fig. 4b).Therefore, after 11:00 in type1, the aerosol in the boundary layer begins to increase while after 12:00in type2, there is an increase in the aerosol in the boundary layer. As shown in Fig. 12, there is a strong southerly wind in type1. Pollutants concentrate in the plain areas along the Taihang and Yan mountains. The $PM_{2.5}$ concentration in the eastern part of the Beijing-Tianjin-Hebei plain was significantly lower than that in the western part along the mountains in type2.Northerly air flow mainly influences the entire atmosphere (above 500m) of type3 in Fig. 10. In general, the air flow in the atmosphere indicates the arrival of cold air, which generally corresponds to good diffusion conditions. However, the lower part of the boundary layer is often associated with a slow wind speed or southerly wind, which indicates that the northerly wind does not reach the ground. This is an important feature of the type3 boundary layer structure. Weak subsidence caused by the

northerly wind restrains the development of the height of the boundary layer, and as a result, aerosols are confined in the boundary layer and cannot spread to high altitudes. The near-surface layer is convergent and ascending, where the convergence of air currents causes the pollutants in the surrounding area to accumulate locally. As shown in Fig. 11, in type3, the wind speed in the Hebei plain area is relatively low, but the northerly surface wind speed in the western and northern mountainous areas of the plain is relatively high. This indicates that there is a northerly wind (Fig. 10, type3) in the upper part of the small wind layer in the Beijing-Tianjin-Hebei plain. The pollutant concentrations in the surface layer of the Beijing-Tianjin-Hebei plain are higher than those in type1 and type2. The boundary layer height in type4 (Fig. 11) is the highest among the four types (Figs. 8e and 10b). The capacity in the boundary layer for aerosols is larger than that of the other three types. The wind speed above 1,500m is weaker, the wind direction below 1,500m is westerly, and the wind speed below 1,500m is smaller, such that there was no significant wind speed in the region, which indicates that there was no strong weather system in the region. From a wind direction perspective, the wind was southerly during the day and northerly at night. This is a typical mountain–plain wind in the Beijing area (Fig. 6).With changes in the mountain and plain winds, there will be a convergence line in the Beijing plain area, which can be occasionally continuous or fractured.

## 4Conclusion

In this study, we objectively classified pollution weather events based on the REOF method using integrated observation data from meteorology and the environment, combined with the ERA-Interim reanalysis data(0.125°×0.125°).We then synthesized the thermodynamic and dynamic structures of the boundary layer under the different pollution weather types to construct the corresponding boundary layer conceptual models. The results show that four weather types mainly affect the pollution events in Beijing: (a) type1: southerly transport,(b) type2: easterly convergence,(c) type3: sinking compression, and (d) type4: local accumulation. The explained variance in the four pollution weather types were 43.69(type1), 33.68 (type2), 16.51(type3), and 3.92% (type4), respectively.

In persistent pollution events, the proportion of type1 and type2 occurrences were more than 80% on the first and second days, with subsequent alternations in the other types. The atmospheric structures of type1, type2, and type3 have typical baroclinic characteristics in the mid-high latitudes, indicating that synoptic-scale systems, together with local circulation, affect the accumulation and transport of pollutants in the boundary layer. On the other hand, the atmospheric structures of type4 have typical barotropic characteristics, which indicates that local circulation plays a major role in pollutant accumulation and transport. This is the first time that the baroclinic and barotropic characteristics of the atmosphere have been introduced into the discussion of pollution boundary layer.

Among the four types, southerly winds, with certain thicknesses and intensities, appeared in the boundary layer of type1, which was favorable for the transportation of pollutants to Beijing, accumulating more in areas along the Yan and Taihang Mountains. On the other hand, the pollution level in the central plain area of Hebei was relatively small. For type2, the pollutants mainly concentrated along the Taihang Mountains due to the influence of the interaction between weak easterly

winds and topography. The vertical structure of the atmosphere was unfavorable for pollutants to ascend into the mountains. Type 3 had the lowest inversion height, boundary layer height, and the highest relative surface humidity, which are favorable for $PM_{2.5}$ hygroscopic growth. Finally, type3 had the highest $PM_{2.5}$ concentration. Type4 had the highest boundary layer height and lowest relative humidity among the four pollution types, whose $PM_{2.5}$ concentration was relatively low when exposed to local mountain–plain winds. Pollutant accumulation is related to dynamic oscillation along the convergence line of the mountain terrain. The results of this study allow us to understand the formation mechanism of different heavy pollution boundary layers from synoptic scale and boundary layer perspectives, as well as to provide scientific support for scientific emissions reduction and air quality prediction. The different heavy pollution weather types and heavy pollution boundary layers not only reflect the interaction between the atmospheric mean flow and fluctuation, but also reflect the process of heavy pollution weather types shaping the boundary layer. Changes in pollution weather patterns cause the pollution boundary layer to change to another type.

Although we attempted to collect data on all types of atmospheric pollution boundary layer structures in the Beijing area, there are still certain data samples that were not collected. These data can also explain the pollution characteristics associated with the four heavy pollution boundary layers from other factors, such as $PM_{2.5}$ composition data. We also speculate that there is feedback between aerosols and the boundary layer, which was not examined in this study. Although there have been numerous studies on atmospheric pollutant transport, there are few studies on 3-Dpollutant transportation, which will be the focus of our future investigations.

*Data availability*. All the data are available upon request via email: xjzhao@ium.cn.

*Competing interests*. The authors declare that they have no conflict of interest.

*Acknowledgements*. This study is supported by the National Natural Science Foundation of China(41305130), Beijing Major Science and Technology Project (Z181100005418014), the Natural Science Foundation of Beijing Municipality (8161004) and the National Natural Science Foundation of China(41975004).

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

## Figures and figure captions

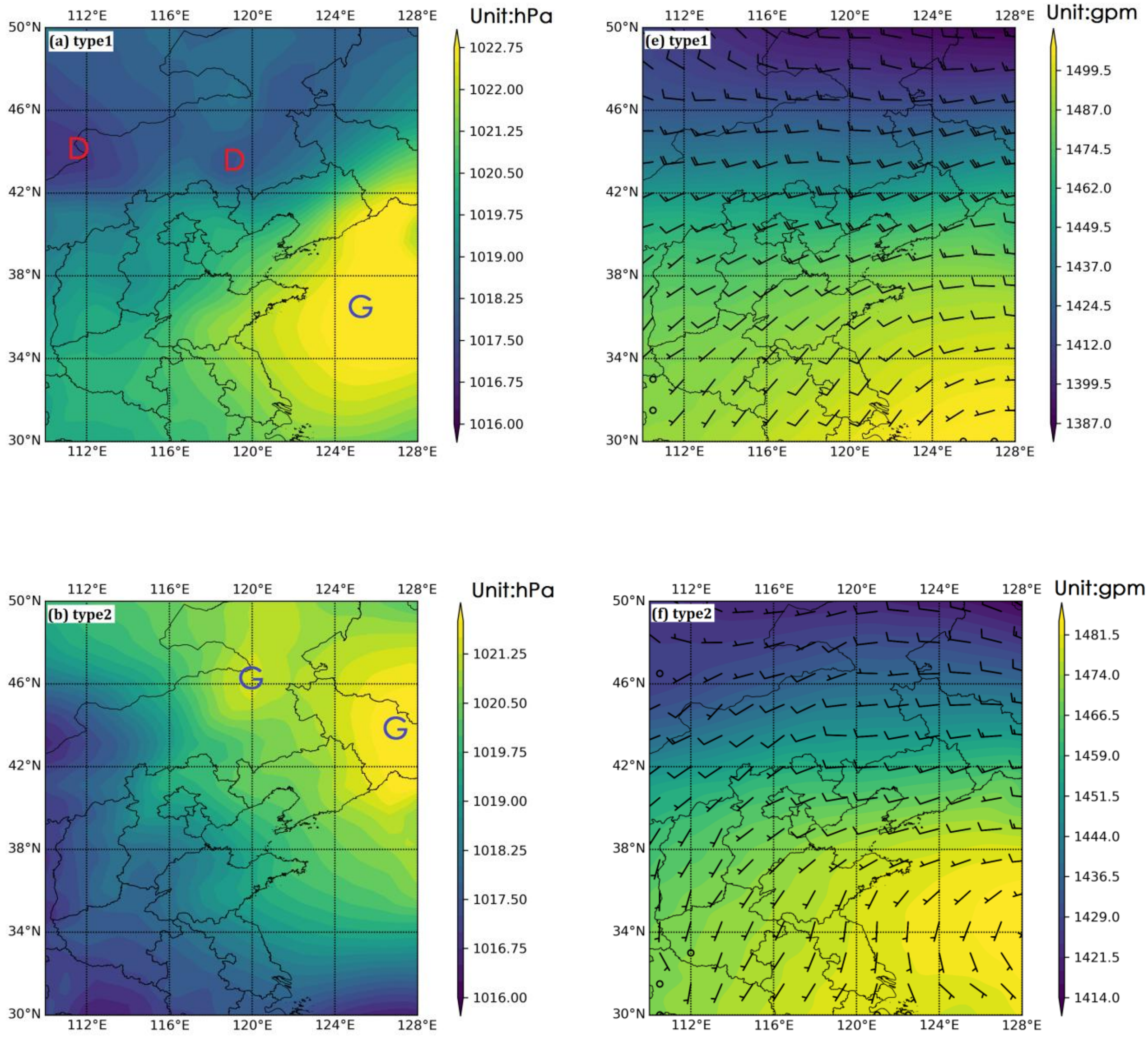

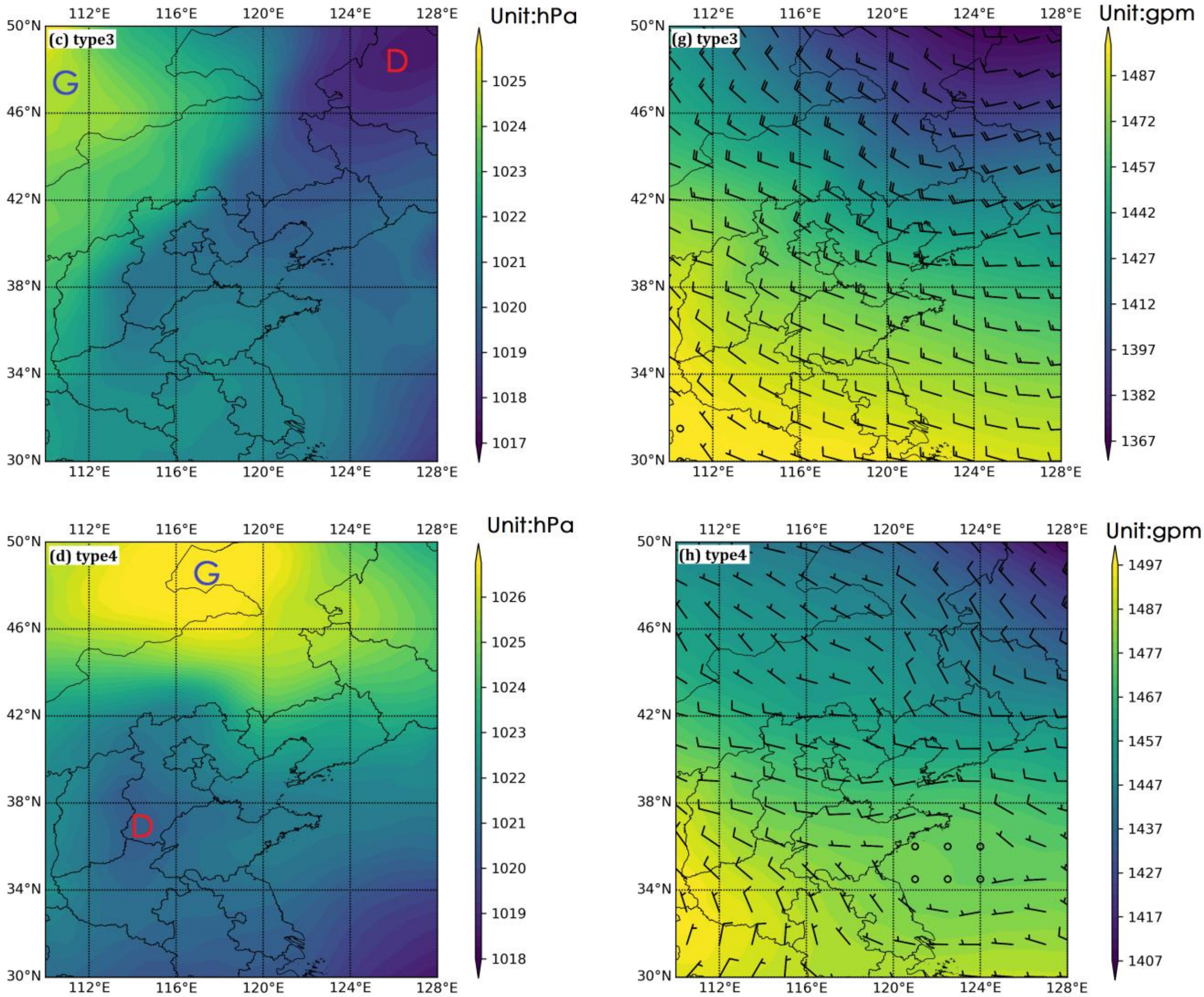

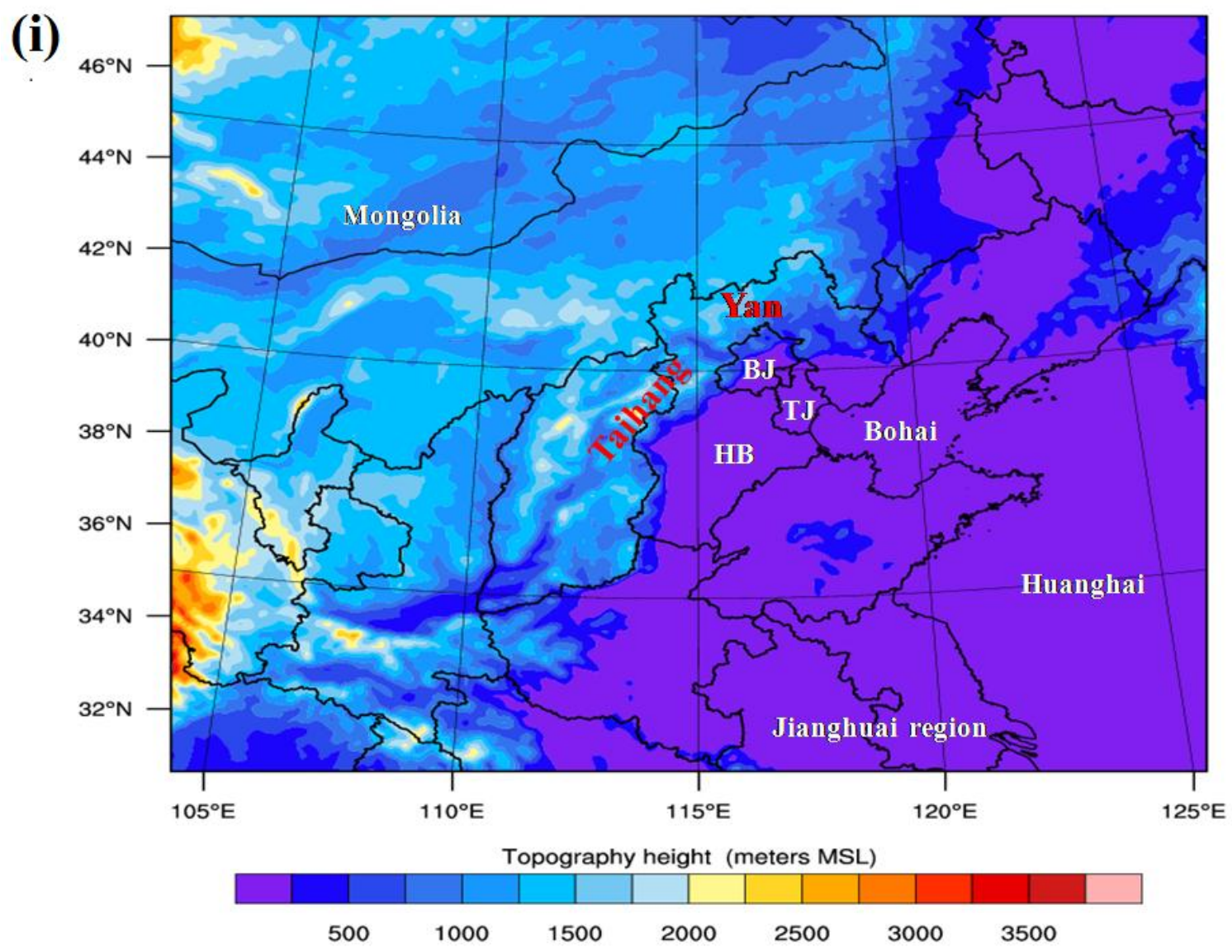

Figure 1. Sea level pressure (unit: hPa, top), geopotential height of 925hPa(unit: gpm, bottom), wind field (wind direction bar)for the four heavy pollution weather types in the Beijing area: (a and e) type1, (b and f) type2, (c and g) type3, and (d and h) type4.BJ,TJ and HB represent Beijing, Tianjin and Hebei. Yan and Taihang represent Yan and Taihang montains.

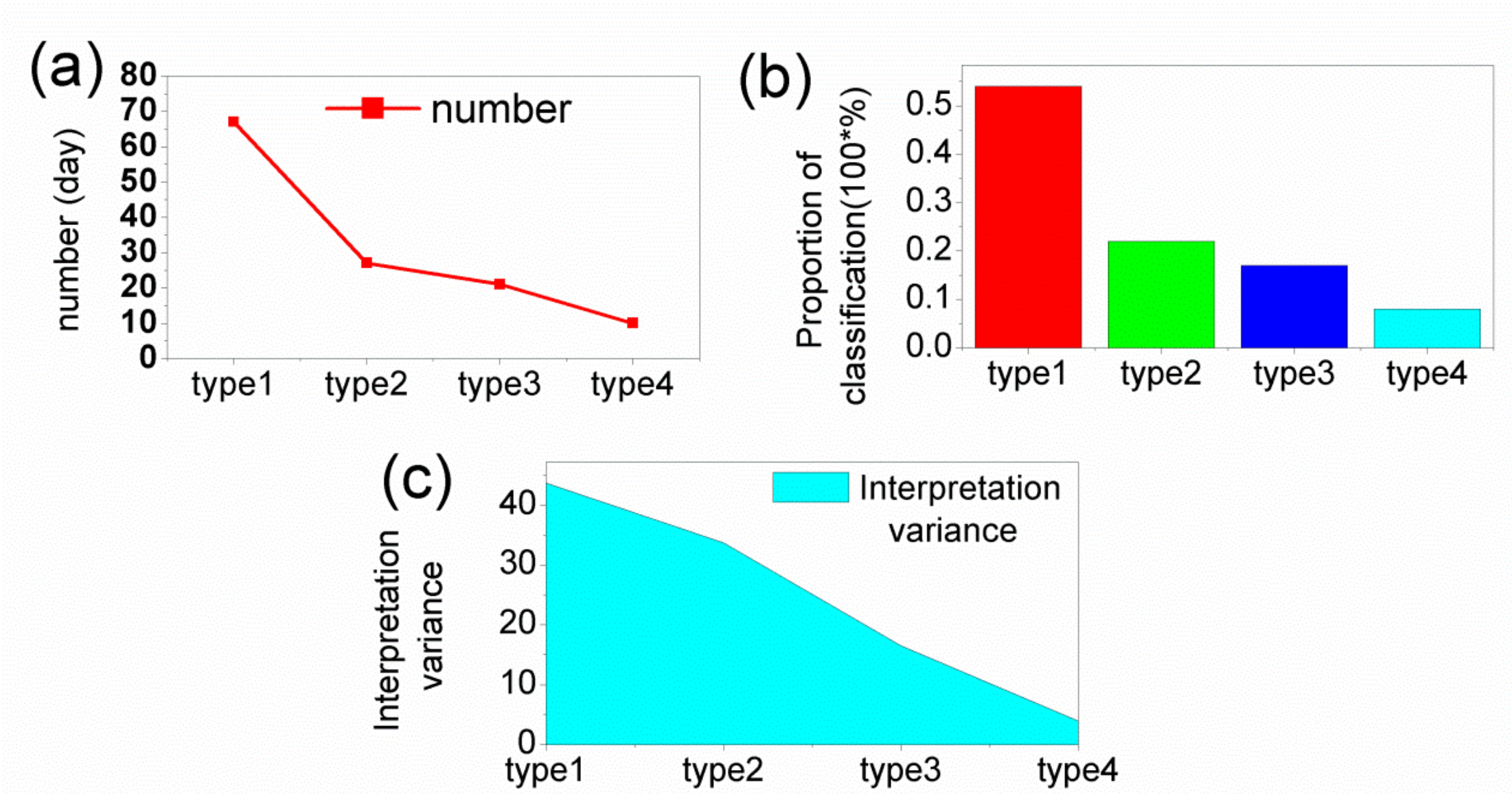

Figure 2.The four pollution weather types as a function of their (a) number of samples,(b) proportion with respect to the total number of samples, and (c)interpretation variance.

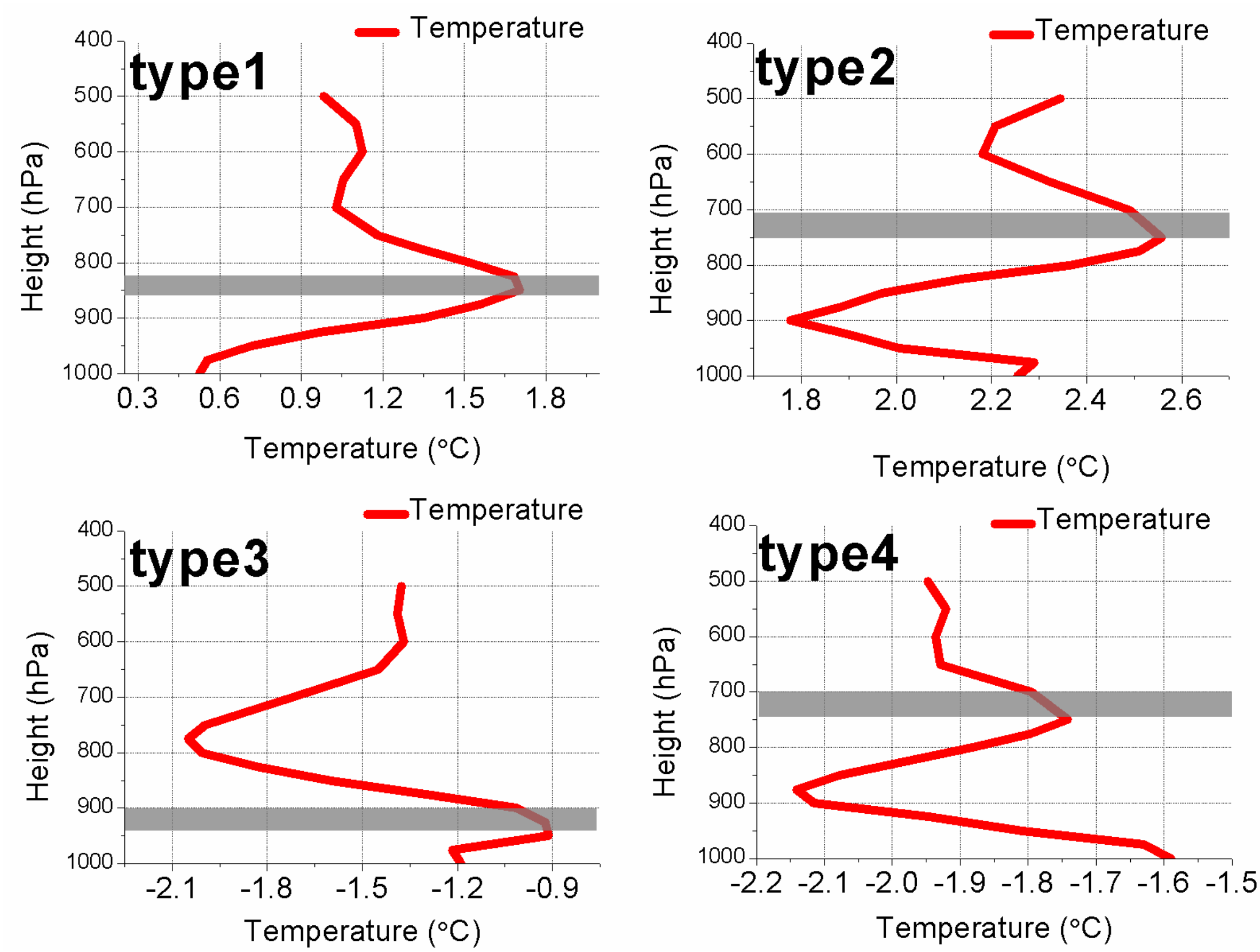

Figure. 3 Vertical distribution of temperature in pollution boundary layer of four types in Beijing area .Solid red lines represent temperatures at different heights. Gray shade represents the top of the inversion layer.

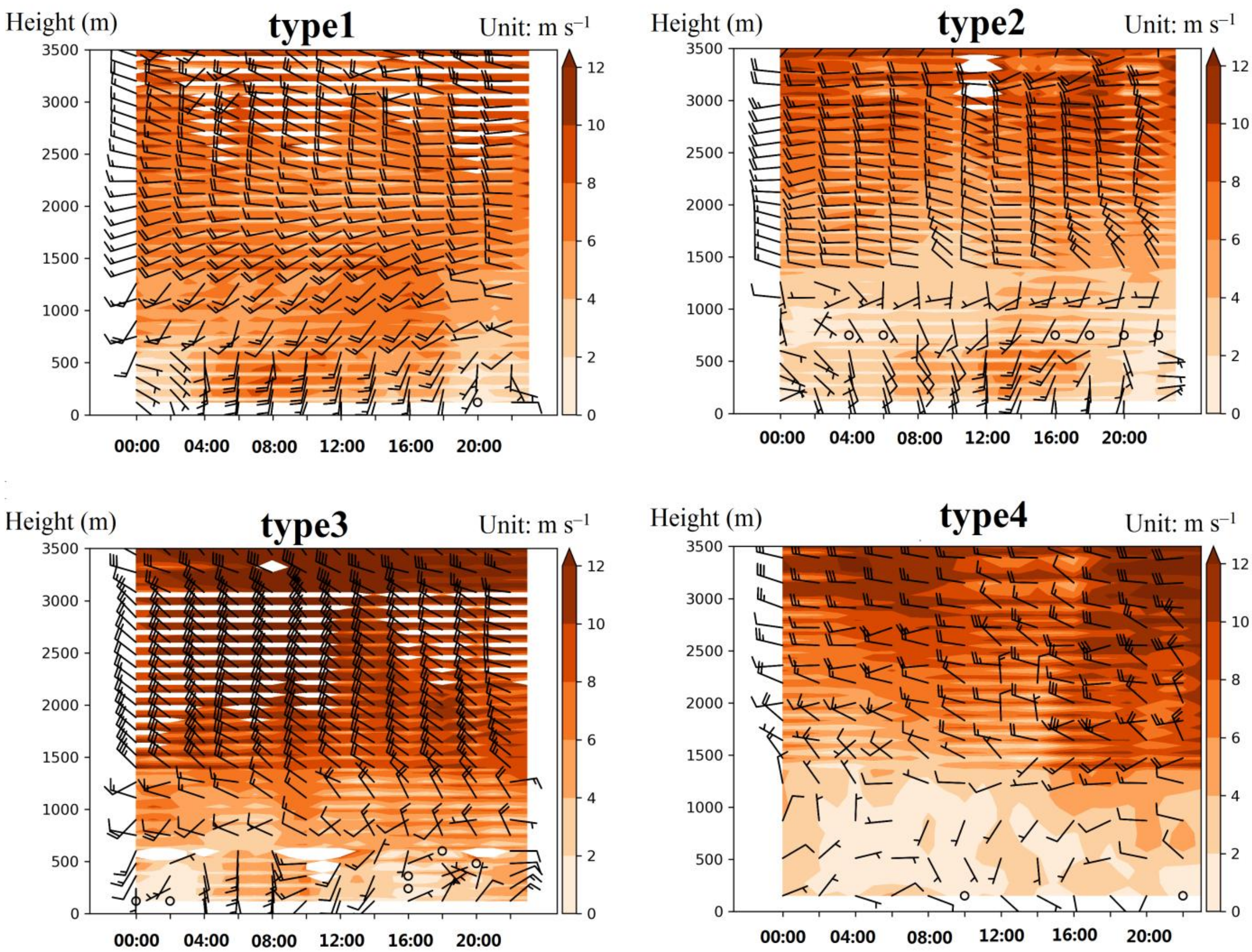

Figure 4.The mean wind field characteristics of the four pollution weather types in the Beijing area (varying colors, based on the color bar to the right of each panel, represent the wind speed in m $s^{-1}$; the x-axis is in Beijing time from 00:00 to 23:00; the y-axis is the height in m).

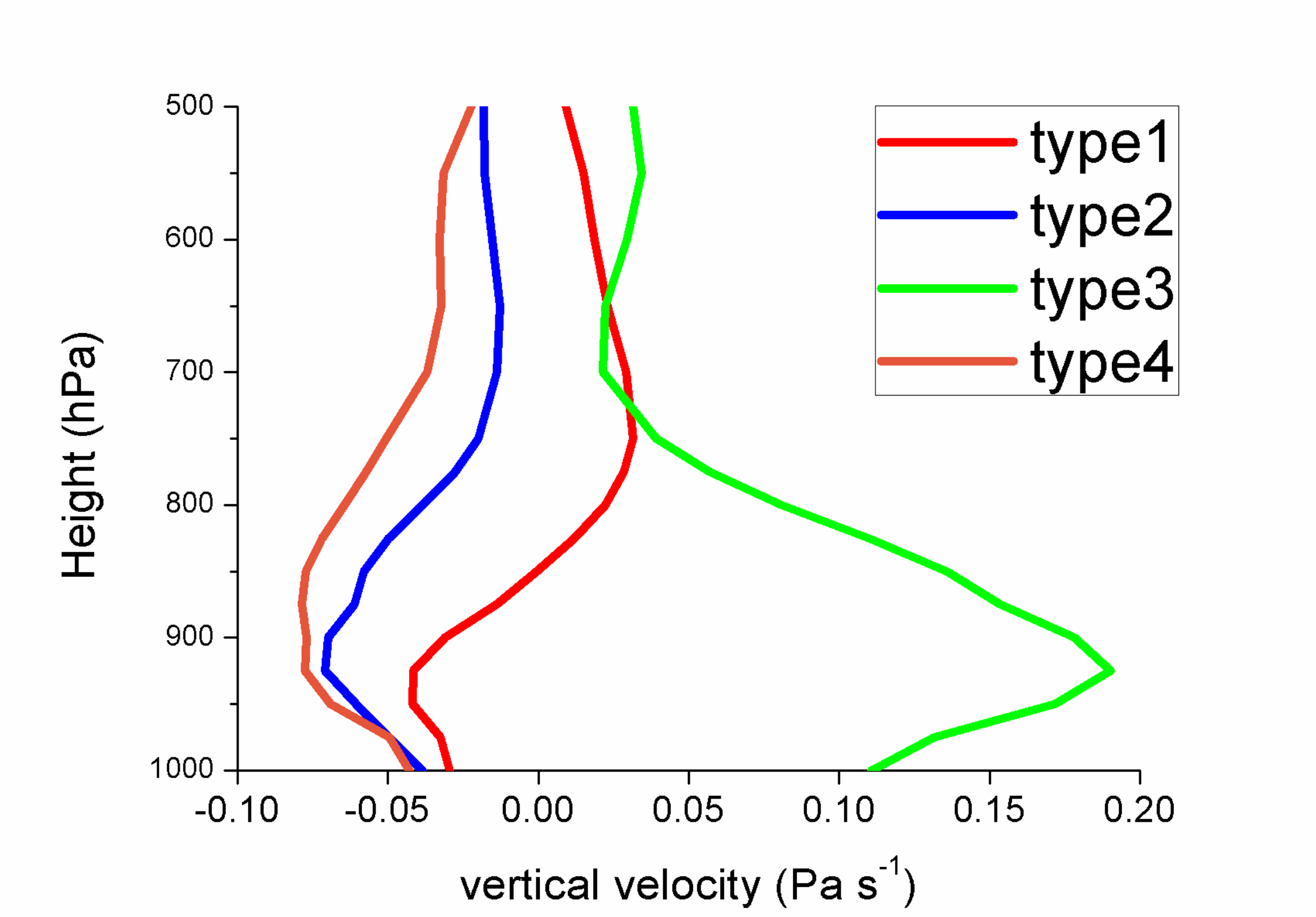

Figure 5.The vertical speed profiles in the four pollution weather types (type1: red, type2: blue, type3: green, and type4: red) in the Beijing area. The negative values represent ascending motion while positive values represent descending motion under the P coordinate(unit: Pa $s^{-1}$).

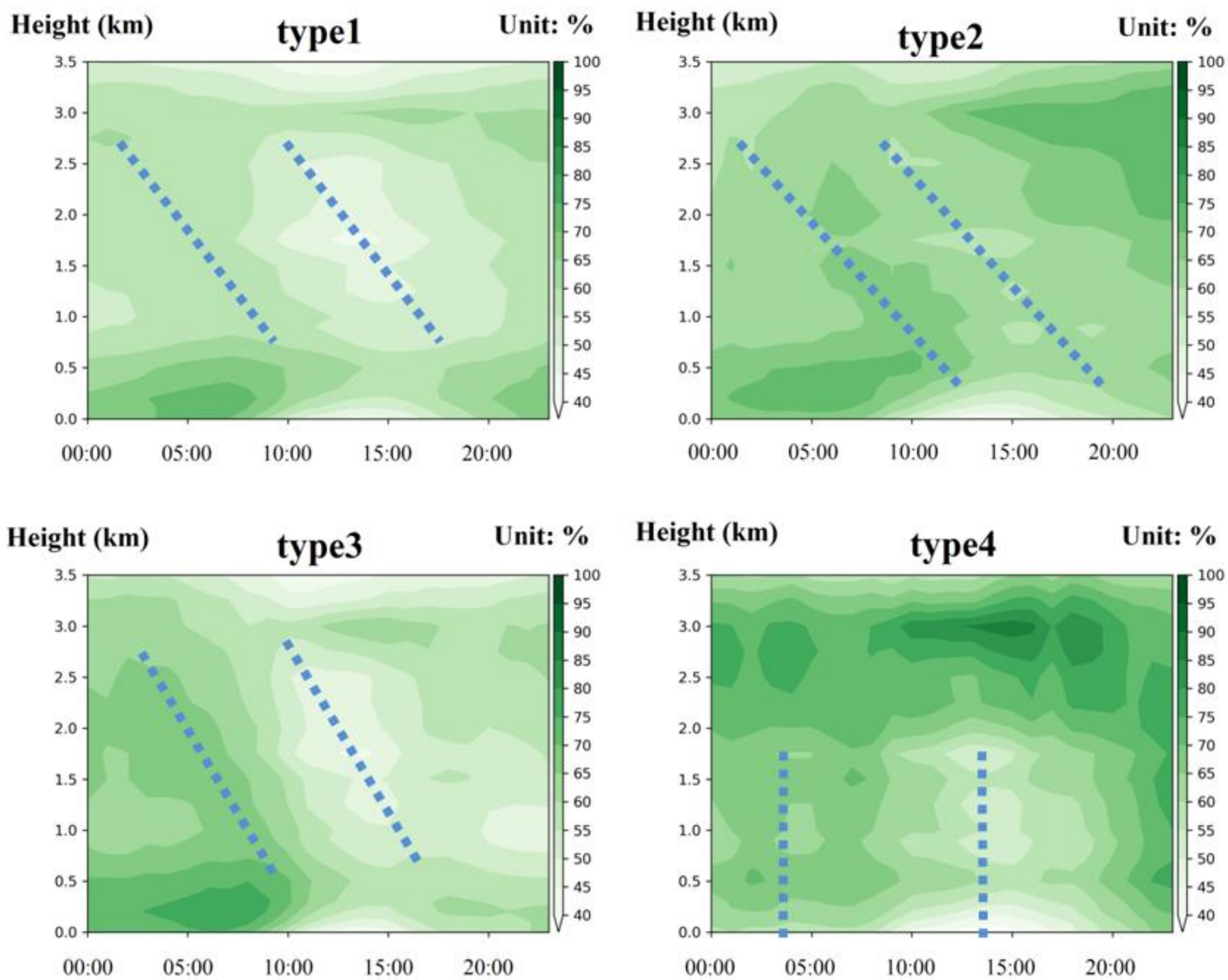

Figure. 6 Average characteristics of the relative humidity field in the boundary layer under four pollution types in Beijing area (shadow represents relative humidity, unit:%; x-axis is Beijing time, from 00:00 to 23:00; y-axis is height, unit: km).

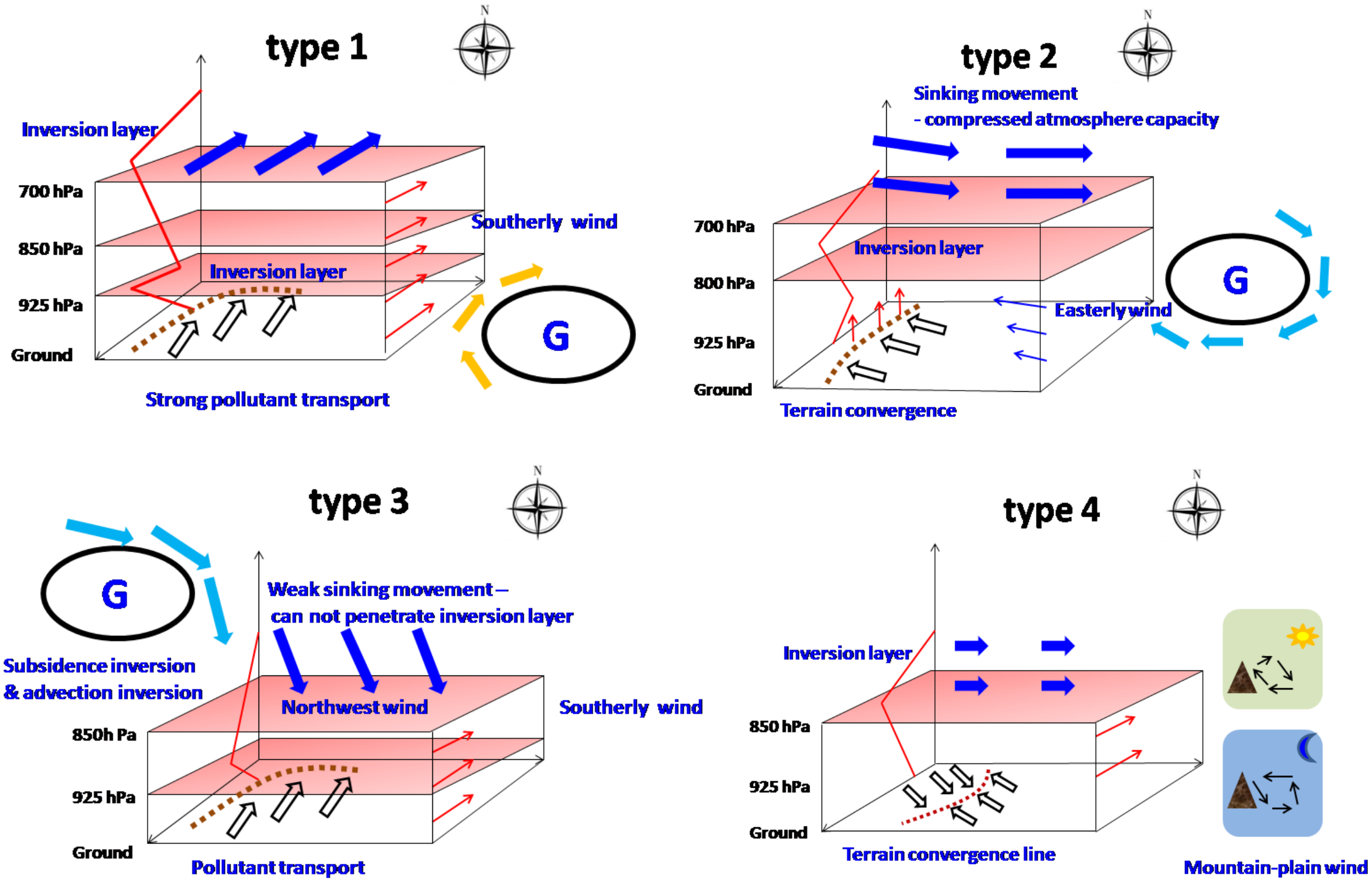

Figure 7.A thermodynamic and dynamic structure conceptual model of the pollution boundary layer for the four types of weather in the Beijing area. Arrows represent wind directions at different heights. Hollow arrow represents the ground horizontal wind field, thin red and blue arrows represent wind fields at different heights, and thick blue arrow represents the upper wind field. The dark red dots represent ground convergence lines, including 1) convergence between wind fields and 2) convergence between wind fields and topography. Solid red line is temperature. The Beijing area is located within the lowest rectangle, and the small figure in type4 represents mountain–plain winds with a daily cycle

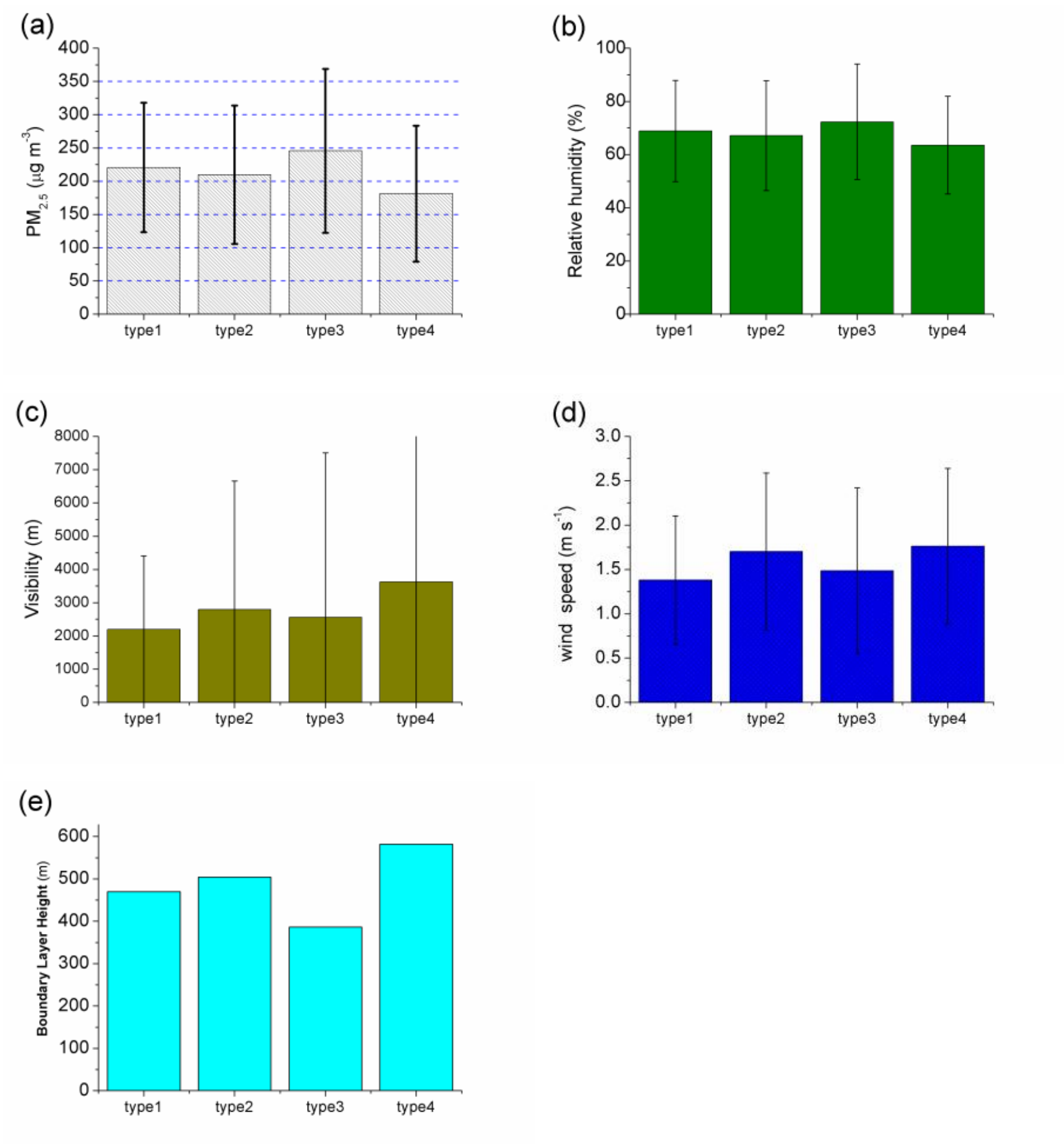

Figure 8. The four pollution weather types in Beijing area: (a) average daily $PM_{2.5}$concentration at 12 state-controlled stations (unit: μgm$^3$), (b) average daily relative humidity at the Beijing Observatory (unit:%), (c) average daily visibility at the Beijing Observatory (unit: m), (d) average daily wind speed at the Beijing Observatory (m s$^{-1}$), and (e) the boundary layer height from the tower station at the Institute of Atmospheric Physics, Chinese Academy of Sciences (unit: m).

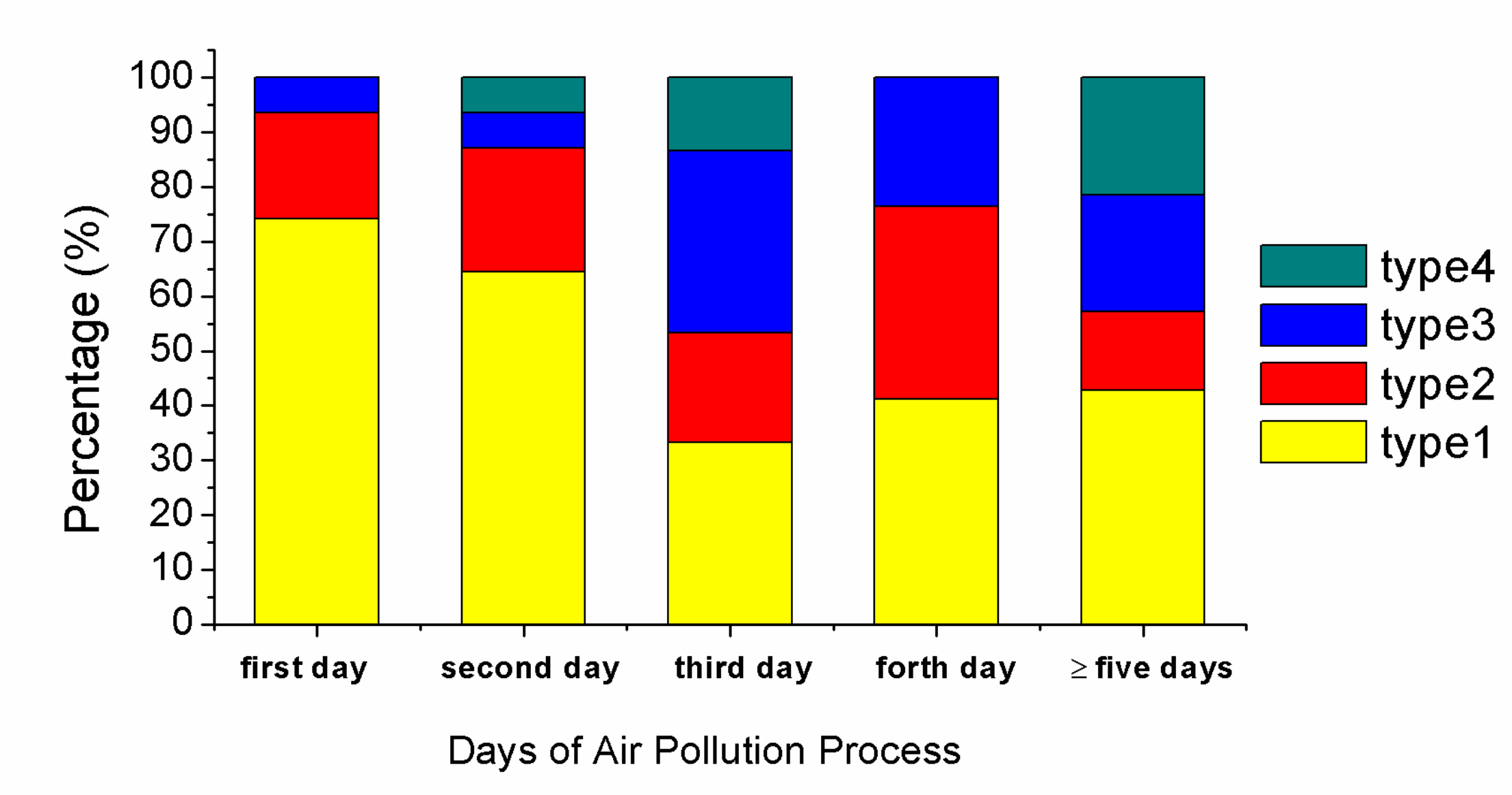

Figure 9. Time distribution of the four pollution weather types (yellow, red, blue, and green represent type1,type2,type3, and type4, respectively) during pollution events in the Beijing area.

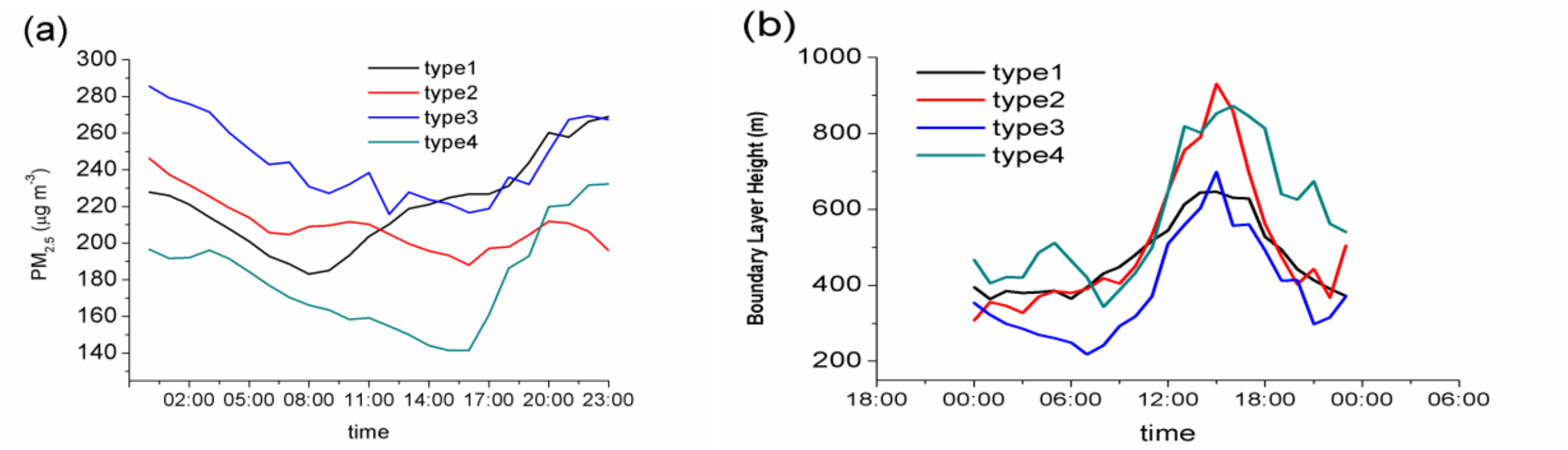

Figure 10. Diurnal variation characteristics of the (a) $PM_{2.5}$ concentration (μg $m^{-3}$) and (b) boundary layer height (m) under the four pollution weather types in the Beijing area (x-axis: 00:00–23:00Beijing time).

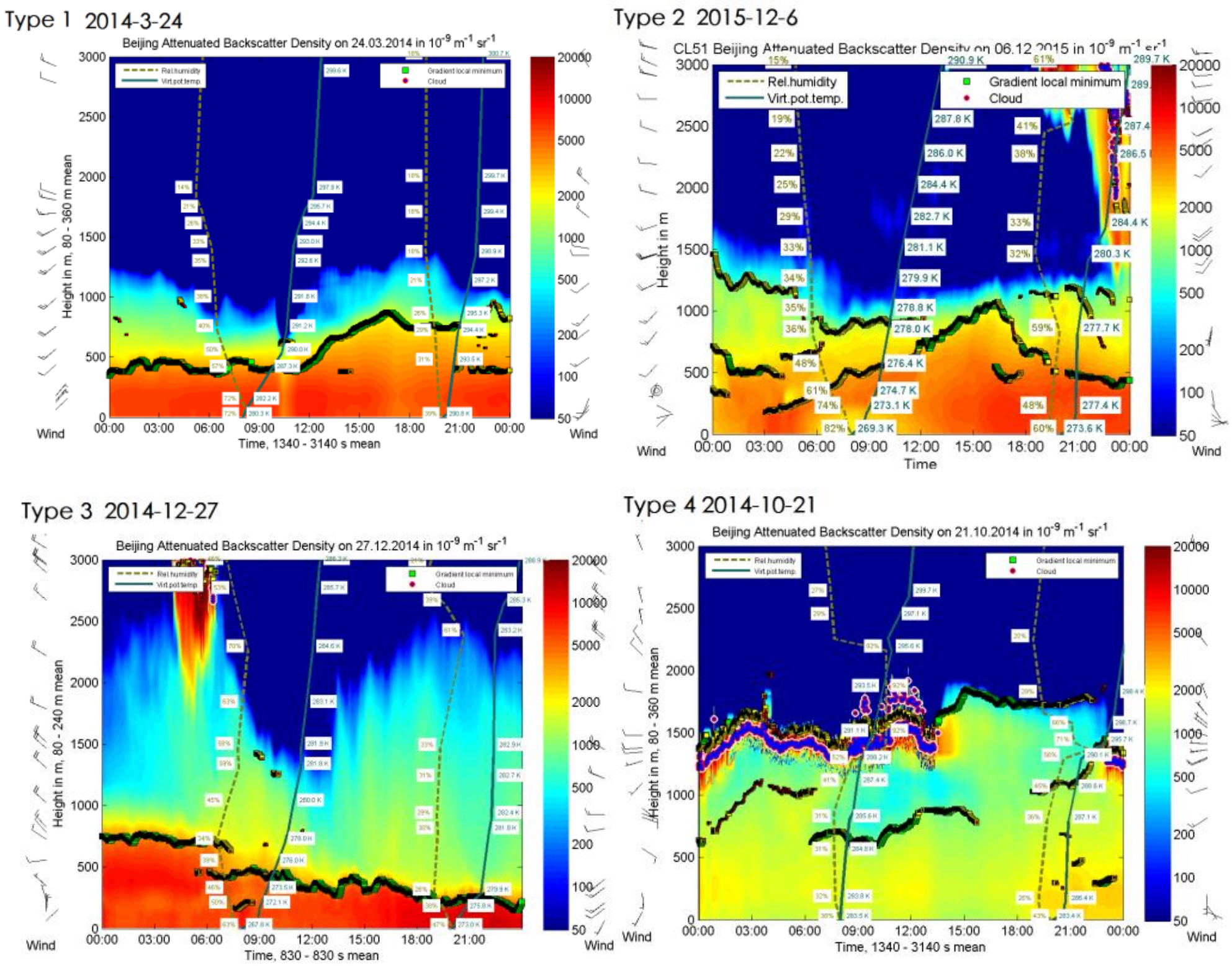

Figure 11. Aerosol backscattering intensity of the four pollution weather types in the Beijing area and the vertical structure of meteorological elements at the Beijing Observatory station (y-axis is height in m and the x-axis is Beijing time from 00:00–23:00).

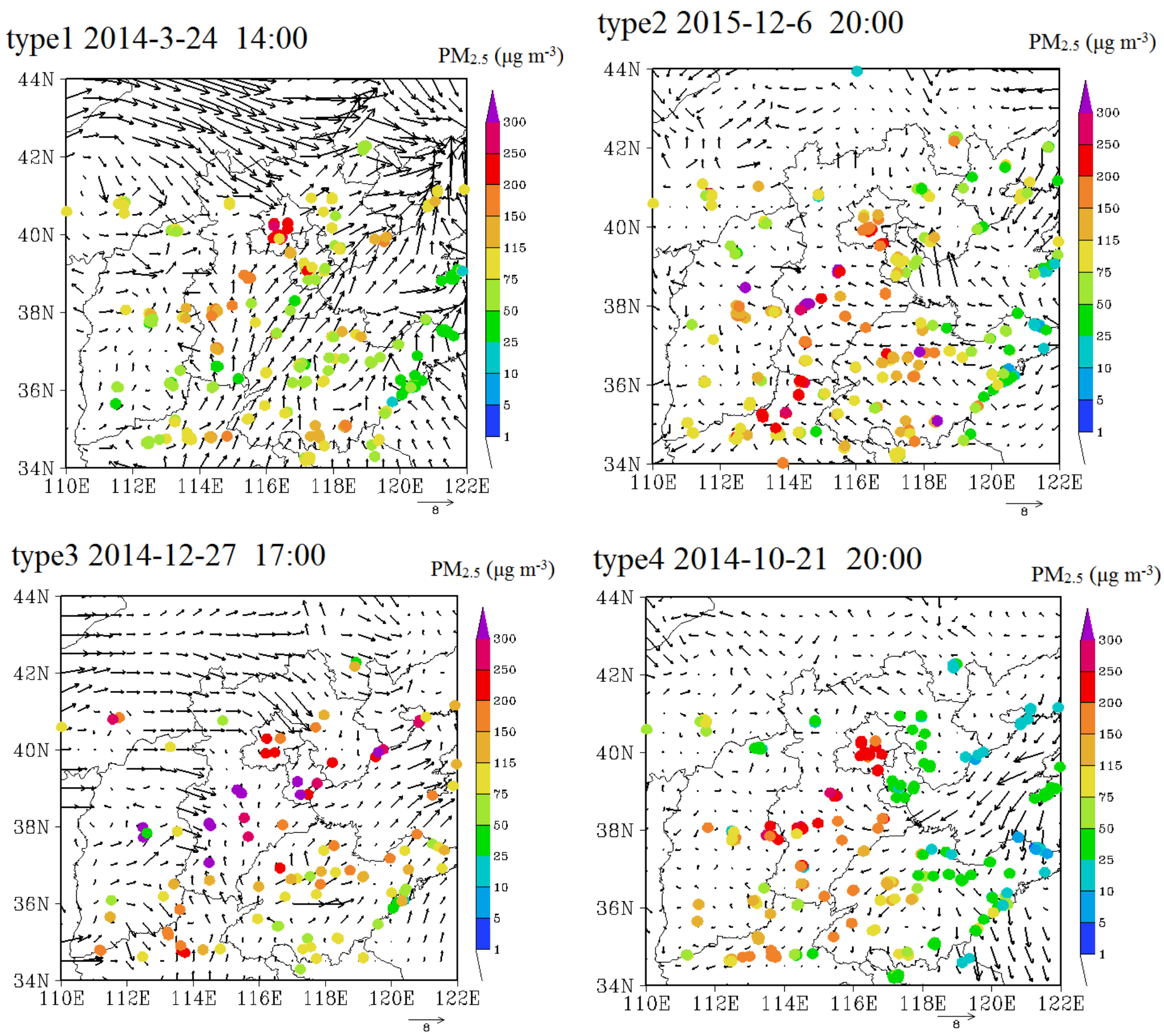

Figure 12. $PM_{2.5}$ concentrations and surface wind fields under the four pollution weather types in the North China. Solid circle represents the air pollutant monitoring stations, different colors represent different levels of pollution.