# Peer review of "Boundary layer structure characteristics under objective classification of persistent pollution weather types in the Beijing area 3"

_Atmospheric Chemistry and Physics, 2020_

## Referee Comment (RC1) · Anonymous Referee #2 · 8 Nov 2020

The paper deals with the pollution in the Beijing area. The authors suggest a method based on objective classification and identify four weather types that can promote intense pollution. The work is interesting, however I have some general concerns. First of all, it is well-know that the pollution is strongly influenced by the meteorological conditions both at local and large scale. In this sense, the paper does not add any novelty to the literature. Second, the number of cases considered (32) seems to be not sufficient for a statistical analysis. Third, the results are not compared with any other methods in order to assess the feasibility of the proposed procedure.

My general impression is that the great effort made by the authors in terms of measurements and data analysis does not yield to robust and founded results.

Other comments

In the discussion the authors go deep into details but the effect is loss of clarity and some of the statements appear to be conjecture rather than evidence.

In Figure 1 it is not clear what does represent the last panel (i). I suggest to move it to another figure.

Figure 3: Are the profiles a mean over a period?

Figure 5: Same as for Figure 3

How long do the events last and how is the duration defined?

How are conceptual models built? Is some method used or are they simply qualitative?

Lines 428-429 This sentence is not clear. What is the pollution weather pattern and how it causes the pollution boundary type.

In summary, it is not clear what the applicability of this method is. It is not clear whether it can be applied to other cases. In general it seems to me that the conclusions of this work are already well known and this article does not add much more. It is known that the height of the boundary layer is related to pollution as well as other weather factors have a direct influence on the presence of pollutants.

---

## Referee Comment (RC2) · Anonymous Referee #1 · 21 Dec 2020

The manuscript covers the important topic of haze due to air pollution from a major source region in China, Beijing and its surrounding area. This is an issue for the population of this region due to degraded air quality and visibility, but is just as important due to the adverse impacts of the associated pollutants on climate.

The authors have developed, or perhaps refined, the Rotated Empirical Orthogonal Function method to classify 125 days of haze over two years, 2014-16, into four types distinguished by their weather patterns and boundary layer structure. The novelty of their approach, according to the authors, is in the combined analysis of synoptic-scale circulation and the boundary layer structure in the analysis of pollution transport into,

within and out of the source region.

My overall assessment is that this topic is worthy of analysis, but the paper leans very heavily on the authors' prior knowledge of meteorology and presents little information on the haze episodes themselves, and how the data were analyzed. The most relevant data for the subject of haze, i.e., the aerosol backscatter data, and the PM monitor data over the region are presented for just four days representing each of the four types, but it is not clear how they were selected, nor where they fall within the range of temporal averages presented in other plots. Likewise, time-averaged meteorological data are presented with no information about why, how, and over what period the temporal averaging was done.

That leads me to the presentation, which needs a lot of work. Most critically the conceptual model that the authors developed to understand the meteorological categories is described but refers to the wrong figure. Figures have not been checked for visual quality of presentation and completeness. Figure captions are incorrect or incomplete, and some figures (Fig. 1i, for example) are included without explanation of their purpose. The section on results and discussion is not well organized, and the information density makes it virtually unreadable in some segments. There are also innumerable instances of typographical and syntax errors, as well as confusing sentence structure.

My recommendation is to reconsider the paper after major revisions, at a minimum to address the comments I have included in sticky notes in the attached manuscript (as a supplement file).

Please also note the supplement to this comment:
https://acp.copernicus.org/preprints/acp-2020-538/acp-2020-538-RC2-supplement.pdf

———————————

[Figure]

**Supplement:**

[revised manuscript text omitted]
 North China. Solid circle represents the air pollutant monitoring stations, different colors represent different levels of pollution.

---

## Author Comment (AC1) · 4 Apr 2021

The paper deals with the pollution in the Beijing area. The authors suggest a methodbased on objective classification and identify four weather types that can promote intensepollution. The work is interesting, however I have some general concerns. Firstof all, it is well-know that the pollution is strongly influenced by the meteorological conditionsboth at local and large scale. In this sense, the paper does not add any noveltyto the literature. Second, the number of cases considered (32) seems to be not sufficientfor a statistical analysis. Third, the results are not compared with any othermethods in order to assess the feasibility of the proposed procedure.My general impression is that the great effort made by the authors in terms of measurements and data analysis does not yield to robust and founded results.

**Response to comments:**

Thanks a lot for the reviewer's comments. To clearly reveal the scientific meaningfulness of this study, we have done much improvement in figures, content, and also the presentation, grammar etc. There are two purposes in this manuscript: first is to study the relationship between synoptic scale system and pollution boundary layer on the basis of objective classification of heavy pollution weather, and the second is to reveal how synoptic pattern shapes pollution boundary layer and influence the formation of haze. We clarified the two points in the new version and reorganized the introduction to highlight the key points accordingly.

In this paper and we have already discussed in section 2.2, this research uses the atmospheric pollution cases are 32, every process for at least more than 3 days, atmospheric pollution of persistent air pollution process standard has undergone strict control and 125 days were selected. Then, the 925hPa geopotential heights with 6h resolution ( 500 samples : 4 times each day in 125 days) were used to determine which mode the pollution events belong to according to the characteristic values of the different pollution, 3000 hours of observation data, with vertical observation of meteorological data (wind profile data, vertical velocity data, microwave radiometer data) are used for further analysis. According to the atmospheric pollution weather type which these samples belong to, the time and vertical structure of the samples are synthesized..The results of the research carried out under the statistical analysis of large sample data should be stable and reliable. We have provided the relevant information in Section 2.2.

We deleted the content of Section 3.4.2 and rewrote it. Based on an analysis of 125 days (more than 3,000 hours) of data., we chose a five-day continuous pollution process to show more details, and also to illustrate the three-dimensional spatiotemporal variation of haze process. We have compared the four types of this study with the existing similar studies in Beijing or North China in the new version.

**Other comments**

In the discussion the authors go deep into details but the effect is loss of clarity and some of the statements appear to be conjecture rather than evidence.

**Response to comments:**

We deleted the content of Section 3.4.2 and rewrote it. At the same time, the other contents of the discussion section have been modified a lot.

In Figure 1 it is not clear what does represent the last panel (i). I suggest to move it to another figure.

**Response to comments:**

Figure 1i represents the topographic height of North China, which is illustrated at the bottom of the figure: Topography height. However, we omit the description in Figure 1, which has caused confusion to reviewer. We are very sorry for that. We have added the instructions.

**Response to comments:**

Sorry for this mistake, we didn't write that we averaged the observed elements in time and space based on classification of weather type. In Section 3.2, related content has been added, as follows:

" Figure 3 to Figure 6 presented the vertical distribution of temperature, wind and RH, respectively. The temperature in Fig. 3 and vertical speed profiles in Fig.5 are averaged for each weather type by using the 6h ERA-Interim reanalysis data, respectively. The mean wind profiles in Fig.4 are observed with a wind profiler radar at the Beijing Observatory, and the mean relative humidity in Fig.6 is measured with a microwave radiometer at the same site. According to the classification of weather type, the spatial-temporal average is carried out."

**How long do the events last and how is the duration defined?**

**Response to comments:**

**It is explained in Section 3.2 of the original text.**

"As this study focuses on episodes of heavy haze pollution, we first defined the criteria. Haze is defined by the relative humidity and visibility. Considering that haze pollution mainly refers to reduced visibility caused by fine particulate matter, as well as taking into account the effects of the pollution levels and duration, the screening criteria for heavy haze pollution were still based on the AQI, PM2.5 concentration, and the duration of low visibility. The specific criteria of a haze pollution event can be defined as follows: the AQI reaches a moderate pollution level (AQI≥150) for more than or equal to 3 days in which at least 1 day reaches the heavy pollution level (AQI>200). The primary pollutant is

PM2.5 in Beijing area. As defined by the AQI, the 24-h average concentration of PM2.5must be above 115  $\mu$ g m-3 for more than three consecutive days and above 150  $\mu$ g m–3 for at least 1 day. At the same time, the accumulated time of horizontal visibility, that is, less than 5 km, has a duration of at least 12 h each day at the Beijing Observatory station.

Based on these criteria, 32 events (125 days) were screened for heavy haze pollution in Beijing between2014 and2016. Eight events occurred in spring and summer while 24 events were concentrated in autumn and winter, 32 events accounting for 75% of the events that occurred during the study period (2014–2016). We collected ground-based routine meteorological observation data in North China, L-band radar second-order sounding data (including wind, temperature, and humidity), wind profile data, ceilometer data, and tower data during these events."

**How are conceptual models built? Is some method used or are they simply qualitative? **Response to comments:**

The establishment of the conceptual model is based on the basic principles of weather dynamics with the comprehensive analysis of the characteristics of weather types and vertical thermal and dynamic structures. Firstly, the daily air pollution weather pattern is objectively classified, and then the vertical structure of horizontal wind, vertical wind, temperature and relative humidity of each weather pattern is calculated by using the spatial-temporal average analysis method. Finally, the characteristics of horizontal and vertical structure of the atmosphere are comprehensively extracted and the conceptual model is established. In Beijing area, 700hPa,850hPa and 925hPaare generally located at a height of about 3000m,1500m and 800m.

Lines 428-429 This sentence is not clear. What is the pollution weather pattern andhow it causes the pollution boundary type.

**Response to comments:**

This description is not accurate and we have deleted it.

In summary, it is not clear what the applicability of this method is. It is not clear whetherit can be applied to other cases.

**Response to comments:**

In the discussion, we try to compare the research results with those in Beijing or North China. In addition, the method is the objective in this study, especially based on REOF weather classification, can be used in other cities and regional, the classification results are reliable, from the ground and upper air meteorological elements analysis, and 3-d structure analysis of aerosol can be seen, the research content is self-consistent.

In general it seems to me that the conclusions of thiswork are already well known and this article does not add much more. It is known that he height of the boundary layer is related to pollution as well as other weather factorshave a direct influence on the presence of pollutants. **Response to comments:**

The innovation of this paper has been explained in the previous section. Our research focuses on the shaping effect of weather patterns on the pollution boundary layer. In addition, the distribution of weather patterns in the process of persistent haze was also studied, which was not carried out by

previous studies. Other cities and regions can also obtain objective pollution boundary layer structure by using the method in this study. Rather than subjective summaries and guesses, to build conceptual models. This greatly saves the manpower cost in the process of air pollution analysis.

---

## Author Comment (AC3) · 4 Apr 2021

**AnonymousReferee #1**

The manuscript covers the important topic of haze due to air pollution from a majorsource region in China, Beijing and its surrounding area. This is an issue for the populationof this region due to degraded air quality and visibility, but is just as importantdue to the adverse impacts of the associated pollutants on climate.

The authors have developed, or perhaps refined, the Rotated Empirical OrthogonalFunction method to classify 125 days of haze over two years, 2014-16, into four typesdistinguished by their weather patterns and boundary layer structure. The novelty oftheir approach, according to the authors, is in the combined analysis of synoptic-scalecirculation and the boundary layer structure in the analysis of pollution transport into,within and out of the source region.

My overall assessment is that this topic is worthy of analysis, but the paper leans veryheavily on the authors' prior knowledge of meteorology and presents little informationon the haze episodes themselves, and how the data were analyzed. The most relevantdata for the subject of haze, i.e., the aerosol backscatter data, and the PM monitor dataover the region are presented for just four days representing each of the four types, butit is not clear how they were selected, nor where they fall within the range of temporalaverages presented in other plots. Likewise, time-averaged meteorological dataare presented with no information about why, how, and over what period the temporalaveraging was done.

**Response to comments:**

Thanks a lot for the reviewer's comments. To strengthen the analysis on the haze episodes, we have added the averaged horizontal distribution of $PM_{2.5}$ concentration accompanying with surface wind field for each weather type as figure 11, and addressed the influences of weather types on them. We also selected a typical haze episode and analyzed the influenced of all the four types on its evolution, so we have rewritten section 3.4.2 in a clearer way and rearranged the figures. All the data we used and averaged are introduced with more detail information in the new version.

That leads me to the presentation, which needs a lot of work. Most critically the conceptualmodel that the authors developed to understand the meteorological categoriesis described but refers to the wrong figure. Figures have not been checked for visualquality of presentation and completeness. Figure captions are incorrect or incomplete,and some figures (Fig. 1i, for example) are included without explanation of their purpose.The section on results and discussion is not well organized, and the informationdensity makes it virtually unreadable in some segments. There are also innumerableinstances of typographical and syntax errors, as well as confusing sentence structure.

**Response to comments:**

Sorry for the mistakes and unclear statement. We have done much improvement in figures, content, and also the presentation, grammar etc.

The most relevant changes are in the introduction, section 3.2and section 3.4. In the introduction, we clarified the focus of our study further and the differences from previous works. In section 3.2, we calculated three elements (relative humidity, wind, inversion intensity) to reflect the thermodynamic features in four episodes and rewrote this part with new results. In 3.4, we mainly strengthen the analysis of a typical haze episode to explain the influences of the four weather types on the evolution an formation of this episode in a 3-D view.

My recommendation is to reconsider the paper after major revisions, at a minimum toaddress the comments I have included in sticky notes in the attached manuscript (asa supplement file).

**Response to comments:**

Thanks a lot for the reviewer's comments. All the comments in the notes have been corrected one by one as following:

31 favorable to particulate matter hygroscopic growth; as a result, the type4 has highest $PM_{2.5}$ concentration. In type4, the

Comments to the authors: I think type3 is the category the authors intend here.

Response to comments: According to the previous comments, type4 has been modified to type3.The experts are probably still looking at the last version, and there is a real problem with the last version.

12 Oct 2020

[revised manuscript text omitted]

Comments to the authors: Where?

Response to comments: Articles have been cited about the haze pollution reducing visibility in the Beijing area.

**References**

[1] Ju Li, Zhaobin Sun, Donald H. Lenschow, Mingyu Zhou, Youjun Dou, Zhigang Cheng, Yaoting Wang, and Qingchun LiAtmos. Chem. Phys., 20,15793–15809, https://doi.org/10.5194/acp-20-15793-2020, 2020.

[2] Wang, Y. H., Liu, Z. R., Zhang, J. K., Hu, B., Ji, D. S., Yu, Y. C., and Wang, Y. S.: Aerosol physicochemical properties and implications for visibility during an intense haze episode during winter in Beijing, Atmos. Chem. Phys., 15, 3205–3215, https://doi.org/10.5194/acp-15-3205-2015, 2015.

[3] Luan, T., Guo, X., Guo, L., and Zhang, T.: Quantifying the relationship between PM2.5 concentration, visibility and planetary boundary layer height for long-lasting haze and fog–haze mixed events in Beijing, Atmos. Chem. Phys., 18, 203–225, https://doi.org/10.5194/acp-18-203-2018, 2018.

40 impact on human health. Haze pollution creates health costs for residents (Dockery et al., 1993; McDonnell et al., 2000) and

Comments to the authors: Where? Clarify. The citations here indicate that you're referring to places in the US and elsewhere.

Response to comments: Thanks very much for your suggestion. Haze has an impact on human health. The geographical area concerned in this study is Beijing, located in north China, so we added our team's previous results on the health impact of haze pollution and particulate matter, and the city is Beijing.

61 influence that the monsoon has on the occurrence frequency of different weather patterns and air quality.

Comments to the authors: the air quality associated with those patterns clarify

**Response to comments:**

Thanks for the reviewer's correction. We corrected it as " examining the air quality associated with those patterns clarify"

.

64 plain wind. Even under conditions associated with weaker synoptic scales, these mesoscale systems largely determine the

Comments to the authors: Surely you mean circulations, not scales?

**Response to comments:**

Thanks, the "scales" was changed to "circulations".

**72** have on the concentration of aerosols. A comprehensive analysis of the these two aspects, that is, combining weather

Comments to the authors: The last part of the previous para suggests that these studies    also look at the impacts of aerosols on the mesoscale atmospheric structure and circulation.

Remove "the"

**Response to comments:**

Thanks, corrected.

**84** boundary layer is not entirely identical. The above-mentioned weather classification method does not take into account the

Comments to the authors: Unclear. Identical to what? Among different    haze episodes?

Of Miao et al and Xu et al?

**Response to comments:**

Sorry for the unclear statement. We have rewritten the this part of introduction, and this sentence has been removed in the new version.

100 system(4-D VAR).The model parameters were changed, and the horizontal resolution was enhanced with the use of more

Comments to the authors: Which ones?

**Response to comments:**

"The model parameters were changed, and the horizontal resolution was enhanced with the use of more"

has been modified to

"There are 4 soil moisture layers with the depth of 7cm, 28cm, 100cm and 255cm respectively. The model contains 20 vegetation types, and the land surface parameters change with the change of vegetation types. "

101 satellite and ground-based observations(https://apps.ecmwf.int/datasets/).

Comments to the authors: Pertaining to what?

**Response to comments:** This sentence has been deleted.

**124** pollution level (AQI>200). The primary pollutant is $PM_{2.5}$ in Beijing area. As defined by the AQI, the 24-h average

Comments to the authors: You mean corresponding to these two AQI values? Clarify.

**Response to comments:**

Sorry for the unclear statement. This sentence was corrected as following:

the AQI reaches a moderate pollution level (AQI≥150) for more than or equal to 3 days in which at least 1 day reaches the heavy pollution level (AQI>200).

**142**layer is more uniform than that of the gaseous pollutants, whereas the particle concentration in the boundary layer is

Comments to the authors: Not clear why gas phase pollutant lifetimes are mentioned here.

**Response to comments:**

We rewrote this sentence as following:

has been modified to

"As the lifetime of a particles is long, that is, several days or weeks, whereas the particle concentration in the boundary layer is generally uniform and significantly different from that in the free atmosphere(Lin et al.,2007;Kang et al.,2019)."

**143** significantly different from that in the free atmosphere. By analyzing the backscattering profile of the atmospheric particles,

Comments to the authors: Reference?

**Response to comments:**

Thanks, reference has been added.

**156** weather. In the empirical orthogonal function (EOF) analysis, the first few main components are the focus of the analysis

Comments to the authors: Rewrite this sentence, not readable.

**Response to comments:**

This sentence has been rewritten as following:"The 925 hPa geopotential height field is affected by both synoptic and local circulations, which can simultaneously reflect the variation characteristics of weather system and boundary layer. Thus, the 925hPa geopotential heights of all pollution events were analyzed in this study by using the 6h ERA-Interim reanalysis data. With 500 samples (4 times each day in 125 days), the rotated empirical orthogonal function (REOF) was used to determine which mode the pollution events belong to according to the characteristic values of the different pollution events. "

**168** In this study, the 925hPa geopotential height was used to classify the pollution weather types into four categories with the

Comments to the authors: Lower panels of the figure appear to be geopotential (gpm). If geopotential height is shown in the figure, it should be in meters.

**Response to comments:**

Thanks, we replot this figure and change the unit to meters.

**175** 43.69, 33.68, 16.51, and 3.92%, respectively (Fig. 2). This indicates that an objective weather classification can effectively

Comments to the authors:The way the figure presents these discrete data as a continuous line is misleading. Present the figure as the data allow. If there are additional data points that went into constructing the interpretation variance plot describe where obtained.

The text jumps around between figures! Better organization is needed to provide the description of the figures in a sequential flow. Also it would be a good idea to present these performance metrics in a table instead of lengthy word descriptions.

**Response to comments:**

The picture has been modified.

[Figure]

Figure 2.The four pollution weather types as a function of their (a) number of samples,(b) proportion with respect to the total number of samples, and (c)interpretation variance.

Thanks for the advice. We are very sorry for the problems caused by our failure to match the pictures with the text.In the previous text, Fig. 2 was used uniformly, without distinguishing between Fig. 2a, Fig. 2b and Fig. 2c, which may make the text lack pertinence. Now we have marked the corresponding picture for each sentence.

It has been modified to
"In this study, we observed 125 days of heavy polluted weather.Among these days, type1, type2, type3, and type4 had 67, 27, 21, and 10 days, respectively (Fig.2a), where the four weather types accounted for 53.6, 21.6, 16.8, and 8.0% of the total sampled weather event days, respectively (Fig.2b). The total interpretation variance of the four types for all events was 97.8% while the independent interpretation variance was 43.69, 33.68, 16.51, and 3.92%, respectively (Fig. 2c)."

177 As shown in Fig. 1, the Beijing area is located toward west of the high-pressure system that has its center located in the sea.
Comments to the authors: Briefly explain how the data were sampled to produce this and the subsequent plots. Single-day (which one and how selected) or multi-day averages (over what days)? Which panel of Fig. 1?
**Response to comments:**
"Among 125 days, type1, type2, type3, and type4 had 67, 27, 21, and 10 days, respectively. Multi-day averaged ERA-Interim data was used to produce sea level pressure and geopotential height of925hPa of four types."
The above content has been added.

183 pollutant convergence into the plains along the Taihang Mountains, When type3 appears, the high pressure center was

Comments to the authors: Tense mismatch, rewrite.

**Response to comments:**

Thank you for the reviewer's carefulness. We corrected it as following:

" When type3 appears, the high pressure center is located in the middle of Mongolia, where Beijing is in the front of the weak high pressure system, with a northwest current at 925hPa.However, the wind speed is lower than that affected by strong cold air, because of which it is difficult to penetrate the lower layer of the boundary layer and the wind can only exist in the upper atmosphere of the boundary layer. "

185 at 925hPa(Fig. 1i). However, the wind speed was lower than that affected by strong cold air, because of which it was

Comments to the authors: Wrong label? Fig. 1i seems to be a topo map. The figure caption is missing the description of Fig. 1i.

**Response to comments:**

At the beginning of this paragraph, "As shown in Fig. 1" has been put forward and (Fig. 1i) has been deleted.

188 of Mongolia and southern Hebei province, where there is only a low pressure system with a smaller spatial and temporal

Comments to the authors:Does this refer to Fig 1d? Rewrite this sentence to clearly indicate that western Mongolia is a high pressure region and Southern Hebei province is a low pressure area. As written in the first half of the sentence, they could both be understood to be high pressure regions.

**Response to comments:**

Yes, this refer to Fig 1d.

Following the reviewer's suggestion, this sentence has been rewritten as: When type4 appears, the center of the high pressure system is located further to the northwest of Mongolia, while only a low pressure system is in the southern Hebei province with a smaller spatial and temporal scale

As can be seen from the Figure 1d and 1h, Beijing area is far from the high pressure system, and the low pressure area is not close to Beijing, so there is no obvious weather system affecting Beijing area. Another evidence is that the wind speed at 925hPa is very small, and the 24-hour wind speed in the boundary layer in Figure 4d is also relatively small. Based on the above evidence, we believe that when Type4 appeared, there was no obvious weather system in Beijing, which resulted in a very small pressure gradient and very small wind speed in boundary-layer, which was very conducive to the development of local circulation.

191 which results in a smaller synoptic-scale pressure gradient in Beijing and the surrounding areas(Fig. 1i). Most areas in North

Comments to the authors: Is Fig. 1i   the correct figure displaying the synoptic-scale pressure gradient? What I see is topography height. Is something else intended in referring to this figure?

Also the font and figure size for 1i are inconsistent with the rest of the panels, and there is no text in the

figure caption corresponding to 1i.

**Response to comments:**

It should be Fig.1d and has been corrected.

Fig. 1i has been added.

"Most areas in North China do not have strong weather systems and the average wind speed in the boundary layer is smaller, which is favorable to the formation and maintenance of the local circulation considering the topography in the Beijing area (Fig. 1i)."

203 is different, where the inversion height of this type is between 700 and 800hPa.

Comments to the authors: What is that mechanism? It's been described before, but does this figure help illustrate it further?

**Response to comments:**

The description here is not accurate, so we delete the " The mechanism of the thermal structure".

204 As shown in Fig. 4, the basic flow is the southerly wind below 2,000m in type1, where a southwest wind appears from 500–

Comments to the authors: The geographic map is not visible in the background. Is it the same area as in Figure 1?

**Response to comments:**

The mean wind profiles in Fig.4 are observed with a wind profiler radar at the Beijing Observatory. We have added more detailed information for figure 3 to figure 6 in the first paragraph in the section 3.2.

The vertical structure of the atmosphere is very important for the formation and evolution of extreme haze events. The vertical thermal and dynamic structures of four weather types are investigated in three-dimensional view. Figure 3 to Figure 6 presented the vertical distribution of temperature, wind and RH, respectively. The temperature in Fig. 3 and vertical speed in Fig.5 are averaged for each weather type by using the 6h ERA-Interim reanalysis data, respectively. The mean wind profiles in Fig.4 are observed with a wind profiler radar at the Beijing Observatory, and the mean relative humidity in Fig.6 is measured with a microwave radiometer at the same site.

In 2.1 Meteorological data, there is description of wind profile data.

" A 12-channel (5water channels and 7oxygen channels) microwave radiometer (Radiometrics, Romeoville, IL, U.S.A.) was used to measure the relative humidity and temperature profile in the atmosphere. The microwave radiometer was installed in the Beijing Observatory (39.93°N, 116.28°E) and was calibrated every three months. The wind profiles, including the wind speed and direction between 100 and 5,000m, are measured at the same station by a wind profiler. The wind profiler radar provides a set of profile data every 6min at a detection height of ~12–16km."

This information is also added in the figure captions:

" Figure 4.Observedmean wind field characteristics of the four pollution weather types in the Beijing area(varying colors, based on the color bar to the right of each panel, represent the wind speed in m s$^{-1}$; the x-axis is in Beijing time from 00:00 to 23:00; the y-axis is the height in m)."

Comments to the authors: Below 500m? Above that it's southwesterly flow and at 2000m is clearly from the west.

**Response to comments:**

Yes, As shown in Fig. 4, the basic flow is the southerly wind below 2,000m in type1, where a southwest wind appears from 500–2,000m. The south wind is below 500m between04:00 and 20:00, and the easterly wind appears at other times. The south wind speed below500m is 4-6 ms$^{-1}$ which is higher than the easterly wind(2-4 m s$^{-1}$)

205 2,000m. The southerly wind is below 500m between 04:00 and 20:00, and the easterly wind appears at other times. The

Comments to the authors: State time standard used (Beijing time).

**Response to comments:**

In 2.1 Meteorological data, " In data analysis, Beijing Local Time (BJT) was used." has been added.

Time standard has been added, such as "04:00" has been changed to "04:00 BJT".

206 southerly wind speed at 500m is strong, while the easterly wind is weak. In type2, the basic flow above 1,000m is westerly

Comments to the authors: Better to quantify "strong" and "weak" by providing the ranges of values.

**Response to comments:**

Thanks for the reviewer's suggestion. This sentence has been rewritten as :

The south wind speed below500m is 4-6 m s$^{-1}$which is higher than the easterly wind(2-4 m s$^{-1}$).

213 above 500m originates from the northwest from 04:00 to 14:00. At altitudes below 500 m, the wind is southerly and

Comments to the authors: during these hours...

**Response to comments:**

Thanks, corrected.

218 local circulation and basic airflow). Westerly or weak northerly winds above 1,000m in type4 control the atmosphere, where

Comments to the authors: Synoptic?

**Response to comments:**

"basic airflow" has been changed to " synoptic airflow".

224 Figure 5 shows that, below 700hPa, type1, type2, and type4 are ascending movements. The maximum of the synoptic scale

Comments to the authors: Vertical axis label should be changed from "Height" to "Pressure". Change "below" to "above" accordingly in the text below.

**Response to comments:**

Thanks. The picture and text have been modified.

227 sinking movement increases gradually with decreasing height, where the maximum of the sinking movement appears at 900–

Comments to the authors: pressure, not height.

**Response to comments:**

Thanks, corrected.

230 all types, resulting in type3 characterized by the ==smallest== capacity among the four types.

Comments to the authors: atmospheric capacity

**Response to comments:**

Thanks, corrected.

231 Based on Fig. ==6,== the relative humidity profiles for the four weather types have both similarities and differences in their space-

Comments to the authors:Change right-axis label of the figure to say RH. Define RH in the figure caption.

To be consistent with the text the height axis needs to be in meters, not km. Easiest if label is changed to show a 10^3 scaling,

**Response to comments:**

The figure has been modified.

[Figure]

232 The similarities in the four types are the increased and decreased relative humidity below 1,000m during the night and day, respectively, with a reverse in the relative humidity layer appearing during the day.

Comments to the authors: Does not seem to hold for Type 4 above 2,000 m.

**Response to comments:**

The analysis here is intended to show the inverse humidity at an altitude about 500m for four types. We changed this sentence as:

"The similarities in the four types are the increased and decreased relative humidity below 1,000m during the night and day, respectively, with a reverse in the relative humidity appearing at an altitude about 500m during the day."

234the surface layer decreases daily from 10:00 to 20:00 with an increase in the solar radiation. The thickness of the dry layer in

Comments to the authors: hours Beijing time. Specify this after time references throughout.

**Response to comments:**

Relevant content has been revised.

236 The top of the dry layer is the reverse of the relative humidity layer. Above 1,000m, the relative humidity of the other three types, except type2, decreases significantly during the day

Comments to the authors: Explain what is happening in Type 4 above 2000 m.

**Response to comments:**

As mentioned in section 3.1, in type4, Beijing is located between a high pressure and a low pressure and in the front of the weak frontal zone. Stratus cloud with high stability is located in the front of the weak frontal zone above 2000m, so relative humidity above 2,000m is high.

246 profile in the pollution boundary layer formed under the condition of wave-current interaction in the atmosphere (Fig. 6).

Comments to the authors: Not clear what this reference to the figure indicates.

**Response to comments:**

The Fig. 6 indicated the fluctuation feature of the basic flow and was moved after this sentence. The last sentence is a summary analysis.

247 Type2 has strong westerly characteristics (Fig. 4), which reflects more baroclinic characteristics in the atmospheric vertical

Comments to the authors: At what height(s)?

**Response to comments:**

We added the height in the revised version as following:

Type2 has strong westerly characteristics below 3,000 m (Fig. 4), which reflects more baroclinic characteristics in the atmospheric vertical structure for the westerlies.

259 type4: local accumulation(Fig. 4). When type 1 appears, the Beijing area is located at the rear of the high-pressure system,

Comments to the authors: Should this reference be to Fig. 7? Fig. 4 as mentioned before contains no geographic map, making it difficult to understand references to geographic features in the following text.

**Response to comments:**

Yes, this reference has modified to Fig. 7. Figure 4 is a mean diurnal wind profile observed at the Beijing Observatory, so there is no need to show the terrain, which has already been explained in above answers.

260 consistent with southerly winds throughout the atmosphere, and multilayer inversion occurs in the boundary layer. Under the

Comments to the authors: That is not obvious in panel 1 of Fig. 4!

**Response to comments:**

Maybe we didn't explain it clearly. It can be seen from Type1 in Figure 1 that all the winds in Beijing and surrounding areas have the component of southerly wind. The panel 1 in Fig.4 indicated that the southerly wind is prevailing below about 2000m. To showed clearly, we added Fig.1e andFig.4 in this sentence.

[Figure]

262 the Hebei region have evident regional transport features. When type2 appears, the Beijing area is located at the bottom of

Comments to the authors: Evident from which figure? Fig. 1?

**Response to comments:**

"The air pollutants in the Hebei region have evident regional transport features (Fig. 1)."

263the high-pressure system, where the air above 850hPa is a westerly wind, with easterly winds below 850hPa. Under the

Comments to the authors: As shown in Fig 4, where the vertical axis is in meters, not hPa? In fact, I can find no figure that shows this.

**Response to comments:**

Because the vertical axes are different for the wind profile and temperature profile, we chose the pressure axis, which is widely used in meteorology, to make conceptual model. In Beijing area, 850hPa is generally located at a height of about 1500m, which is explained in this revised version.

[Figure]

"When type2 appears, the Beijing area is located at the bottom of the high-pressure system, where the air above 850hPa (about 1,500m in Beijing) is a westerly wind, with easterly winds below 850hPa. Under the influence of easterly winds below 850hPa, haze pollutants tend to accumulate in front of the Taihang Mountains."

287Under the influence of easterly winds below 850hPa, haze pollutants tend to accumulate in front of the Taihang Mountains.

Comments to the authors:This is not obvious from Fig. 4!

**Response to comments:**

Fig. 4 is the time profile of the wind profile, and terrain cannot be added. But it can be inferred that the north of the Beijing plain is the Yanshan Mountains and the west of the plain is the Taihang Mountains.

The north and west of Beijing are rigid boundaries. In Figure 2, the westerly winds and northwest winds above 1,000m also play a role in preventing the upward diffusion of pollutants below 1,000m.When easterly winds appear within 1,000m, pollutants cannot move to other directions and can only accumulate along the mountain plain.

296 wind speeds of type2 and type4 are relatively faster, that is, 1.70 and 1.76 m s$_{-1}$, respectively (Fig. 8d). There is a significant

Comments to the authors: Winds are "faster". Wind speeds are "higher".

**Response to comments:**

Thanks, corrected.

300 events (Fig. 9). The daily synoptic types from the first to eighth day of persistent haze pollution events were calculated. As

Comments to the authors: Horiz. axis label has a typo (fourth, not forth).

**Response to comments:**

Sorry for the typo, we corrected it in the new figure.

[Figure]

309 day. The timing of the initial rise in the PM$_{2.5}$ concentration is the earliest among the four types, which indicates the role of

Comments to the authors: This is difficult to understand    as stated because so far no temporal trends of PM concentrations have been discussed (Fig. 10). Introduce that figure and refer to it here.

**Response to comments:**

The above confusion may be the reason that we did not draw the distribution of the four types of PM$_{2.5}$ and wind field. Figure 11 has been added into the original text.

[Figure]

[Figure]

[Figure]

Figure 11.Spatial distribution of thePM2.5 concentration and wind fieldnear the ground.

324The change in the trend of the boundary layer height is similar to that type2 and type4 for most of the day.

Comments to the authors: Between type 2 and type 4?

**Response to comments:**

Thanks for the reviewer's reminder. We corrected this sentence as following:

"The change in the trend of the boundary layer height is similar between type 2 and type 4 for most of the day."

[Figure]

336 9). As different types of weather systems form haze pollution events, we discuss the type of boundary layer structure formed

Comments to the authors:in the next section?

**Response to comments:**

Yes, relevant content has been revised.

340 Figure 11shows the aerosol vertical distribution under the influence of the boundary layer structure for the four pollution

Comments to the authors: This figure is hard to read.

Use consistent axis labels in all four panels. What are the means reported in the time axis labels and height axis labels, and why do they change in each panel? Why are they not reported in the type 2 plot (only)?

The embedded labels for Type 2 and Type 3 overflow the plot area.

What is CL51 (Type 2 panel)?

The wind barbs on the left and right axes make this a very busy plot. The vertical axis label omits to include units. The figure caption makes no mention of winds.

Are the dates presented for each type here the same as in other figures? No mention is made of the episode date elsewhere.

The vertical distribution is inferred, not actually shown. The figure actually shows backscatter density (according to the figure caption, which fails to mention the units).

**Response to comments:**

Figures have been redrawn and simplified.In addition, we deleted this part of content and re-selected a pollution process for five consecutive days to carry out analysis.

341weather types. The wind below 2,000m for type1 in Fig. 11 is southerly (Fig. 4), which facilitates regional pollutant transport.

Comments to the authors: See Fig. 4

**Response to comments:**

Figures have been redrawn and simplified. In addition, we deleted this part of content and re-selected a pollution process for five consecutive days to carry out analysis.

357 (Fig. 4b).Therefore, after 11:00 in type1, the aerosol in the boundary layer begins to increase while after 12:00in type2, there

Comments to the authors: That is not how the figure is labeled. Fig. 4 caption contains no reference to 4b.

**Response to comments:**

"which results in the strengthening of southwesterly winds in the boundary layer 356 (Fig. 4b)"

has been changed to

"which results in the strengthening of southwesterly winds in the boundary layer 356 (Fig. 4)"

358 is an increase in the aerosol in the boundary layer[1]. As shown in Fig. 12[2], there is a strong southerly wind[3] in type1. Pollutants

Comments to the authors:

1)This sentence is a bit jumbled. What is the point being made here? That the timing of the increase is different for the two types?

**Response to comments:**

369 mountainous areas of the plain is relatively high. This indicates that there is a northerly wind (Fig. 10, type3) in the upper

Comments to the authors:No geographical information to infer this on Fig. 11! Which figure is being used for this? Fig. 12?

**Response to comments:**

Figures have been redrawn and simplified. In addition, we deleted this part of content and re-selected a pollution process for five consecutive days to carry out analysis.

372 highest among the four types (Figs. 8e and 10b). The capacity in the boundary layer for aerosols is larger than that of the

Comments to the authors: Unnecessary to refer to these, it can be seen from Fig. 11.

**Response to comments:**

Figures have been redrawn and simplified. In addition, we deleted this part of content and re-selected a pollution process for five consecutive days to carry out analysis.

387days, with subsequent alternations in the other types. The atmospheric structures of type1, type2, and type3 have typical

Comments to the authors: To the other types?

**Response to comments:**

Relevant content has been revised.

398 Type 3 had the lowest inversion height, boundary layer height, and the highest relative surface humidity, which are favorable

Comments to the authors: surface relative humidity.

**Response to comments:**

Relevant content has been revised.

399 for $PM_{2.5}$ hygroscopic growth. Finally, type3 had the highest $PM_{2.5}$ concentration. Type4 had the highest boundary layer

Comments to the authors: Correspondingly

**Response to comments:**

Relevant content has been revised.

404 scientific emissions reduction and air quality prediction. The different heavy pollution weather types and heavy pollution

Comments to the authors: Rewrite this statement. No case has been made for the location of sources, the constituent pollutants, or the emissions controls that would be effective, given what you have analyzed. The discussion has been focused entirely on the meteorology, with Beijing as a whole being an emission "source". A greater level of knowledge of source contributions and locations would be needed before this analysis can be applied to air pollution mitigation.

**Response to comments:**

"The results of this study allow us to understand the formation mechanism of different heavy pollution boundary layers from synoptic scale and boundary layer perspectives, as well as to provide scientific support for scientific emissions reduction and air quality prediction."

has been modified to

"The results of this study allow us to understand the formation mechanism of different heavy pollution boundary layers from synoptic scale and boundary layer perspectives."

As the reviewer said, adding a strategy proposal to reduce emissions needs strong support.

---

## Author Response (AR2)

**AnonymousReferee #1**

The manuscript covers the important topic of haze due to air pollution from a majorsource region in China, Beijing and its surrounding area. This is an issue for the population of this region due to degraded air quality and visibility, but is just as important ue to the adverse impacts of the associated pollutants on climate.

The authors have developed, or perhaps refined, the Rotated Empirical OrthogonalFunction method to classify 125 days of haze over two years, 2014-16, into four typesdistinguished by their weather patterns and boundary layer structure. The novelty of their approach, according to the authors, is in the combined analysis of synoptic-scale circulation and the boundary layer structure in the analysis of pollution transport into, within and out of the source region.

My overall assessment is that this topic is worthy of analysis, but the paper leans veryheavily on the authors' prior knowledge of meteorology and presents little informationon the haze episodes themselves, and how the data were analyzed. The most relevantdata for the subject of haze, i.e., the aerosol backscatter data, and the PM monitor dataover the region are presented for just four days representing each of the four types, butit is not clear how they were selected, nor where they fall within the range of temporalaverages presented in other plots. Likewise, time-averaged meteorological dataare presented with no information about why, how, and over what period the temporalaveraging was done.

**Response to comments:**

Thanks a lot for the reviewer's comments. To strengthen the analysis on the haze episodes, we have added the averaged horizontal distribution of  $PM_{2.5}$  concentration accompanying with surface wind field for each weather type as figure 11, and addressed the influences of weather types on them. We also selected a typical haze episode and analyzed the influenced of all the four types on its evolution, so we have rewritten section 3.4.2 in a clearer way and rearranged the figures. All the data we used and averaged are introduced with more detail information in the new version.

That leads me to the presentation, which needs a lot of work. Most critically the conceptualmodel that the authors developed to understand the meteorological categoriesis described but refers to the wrong figure. Figures have not been checked for visualquality of presentation and completeness. Figure captions are incorrect or incomplete, and some figures (Fig. 1i, for example) are included without explanation of their purpose. The section on results and discussion is not well organized, and the informationdensity makes it virtually unreadable in some segments. There are also innumerableinstances of typographical and syntax errors, as well as confusing sentence structure.

**Response to comments:**

Sorry for the mistakes and unclear statement. We have done much improvement in figures, content, and also the presentation, grammar etc.

The most relevant changes are in the introduction, section 3.2 and section 3.4. In the introduction, we clarified the focus of our study further and the differences from previous works. In section 3.2, we calculated three elements (relative humidity, wind, inversion intensity) to reflect the thermodynamic features in four episodes and rewrote this part with new results. In 3.4, we mainly strengthen the analysis of a typical haze episode to explain the influences of the four weather types on the evolution an formation of this episode in a 3-D view.

My recommendation is to reconsider the paper after major revisions, at a minimum toaddress the comments I have included in sticky notes in the attached manuscript (as a supplement file). **Response to comments:**

Thanks a lot for the reviewer's comments. All the comments in the notes have been corrected one by one as following:

31 favorable to particulate matter hygroscopic growth; as a result, the type4 has highest PM2.5 concentration. In type4, the

Comments to the authors: I think type3 is the category the authors intend here.

Response to comments: According to the previous comments, type4 has been modified to type3. The experts are probably still looking at the last version, and there is a real problem with the last version.

12 Oct 2020

This preprint is currently under review for the journal ACP.

Boundary layer structure characteristics under objective classification of persistent pollution weather types in the Beijing area

Zhaobin Sun10, Xiujuan Zhao1, Ziming Li2, Guiqian Tang3, and Shiguang Miao1, Institute of Urban Meteorology, China Meteorological Administration, Beijing 100089. Institute of Urban Meteorology, China Meteorological Administration, Beijing 100089, China Environmental Meteorology Porceast Center of Beijing-Tianin-Hebei, Beijing 100089, China State Key Laboratory of Atmospheric Boundary Layer Physics and Atmospheric Chemistry, Institute of Atmospheric Physics, Chinese Academy of Sciences, Beijing 102300, China

Received: 02 Jun 2020 – Accepted for review: 05 Oct 2020 – Discussion started: 12 Oct 2020

Received: 02 Jun 2020 – Accepted for review: 05 Oct 2020 – Discussion started: 12 Oct 2020 Abstract. Different types of pollution boundary layer structures form via the coupling of different synophic systems and local mesoscale circulation in the boundary layer; this coupling contributes toward the formation and continuation of haze pollution. In this study, we objectively classify the 32 heavy haze pollution events using integrated meteorological and environmental data and ERA-Interim analysis data based on the boundary layer; this coupling contributes toward the formation and continuation of haze pollution. In this study, we objectively classify the 32 heavy haze pollution events using integrated meteorological and environmental data and ERA-Interim analysis data based on the rotated empirical orthogonal function method. The thermodynamic and dynamic structures of the boundary layer for different pollution events types are synthesized, and the corresponding three-dimensional boundary layer conceptual models for haze pollution are constructed. The results show that four weather types mainly influence haze pollution events in the Beiging area: (a) type1: southerly transport, (b) type2: easterly convergence, (c) type3: sinking compression, and (d) type4: local accumulation. The explained haze pollution events, type1 and type2 surpass 80% on the first and second days, while the other types are present alternately in later stages. The atmospheric structures of type4, has typical barotopic characteristics, indicating that the accumulation and transport of pollutants in the boundary layer is affected by couple diructures in synoptic-scale systems and local inculation. The atmospheric structure of type4 has typical barotopic characteristics, indicating that the accumulation, and transport of pollutants is nother areas are relatively low. Due to the interaction between weak easterly winds and the mestry prevail in the boundary layer, which is favorable for the accumulation of pollutants in type2. The at

39 Pollutants not only affect the climate system but also reduce visibility, affect city operation, and have a significant negative

Comments to the authors: Where?

Response to comments: Articles have been cited about the haze pollution reducing visibility in the Beijing area.

**References**

[1] Ju Li, Zhaobin Sun, Donald H. Lenschow, Mingyu Zhou, Youjun Dou, Zhigang Cheng, Yaoting Wang, and Qingchun LiAtmos. Chem. Phys., 20,15793-15809, https://doi.org/10.5194/acp-20-15793-2020, 2020.

[2] Wang, Y. H., Liu, Z. R., Zhang, J. K., Hu, B., Ji, D. S., Yu, Y. C., and Wang, Y. S.: Aerosol physicochemical properties and implications for visibility during an intense haze episode during winter in Beijing, Atmos. Chem. Phys., 15, 3205-3215, https://doi.org/10.5194/acp-15-3205-2015, 2015.

[3] Luan, T., Guo, X., Guo, L., and Zhang, T.: Quantifying the relationship between PM2.5 concentration, visibility and planetary boundary layer height for long-lasting haze and fog-haze mixed events in Beijing, Atmos. Chem. Phys., 18, 203-225, https://doi.org/10.5194/acp-18-203-2018, 2018.

40 impact on human health. Haze pollution creates health costs for residents (Dockery et al., 1993; McDonnell et al., 2000) and

Comments to the authors: Where? Clarify. The citations here indicate that you're referring to places in the US and elsewhere.

Response to comments: Thanks very much for your suggestion. Haze has an impact on human health. The geographical area concerned in this study is Beijing, located in north China, so we added our team's previous results on the health impact of haze pollution and particulate matter, and the city is Beijing.

**References**

[1] Han,L., Sun,Z.B., He,J., Zhang,X.L., Hao,Y., Zhang,Y.: Does the early haze warning policy in Beijing reflect the associated health risks, even for slight haze? Atmos. Environ., 210, 110–119, https://doi.org/10.1016/j.atmosenv.2019.04.051,2019.

[2] Han,L., Sun,Z.B., He,J., Hao,Y., Tang,Q.L., Zhang,X.L.,Zheng,C.J., Miao,S.G.:Seasonal variation in health impacts associated with visibility inBeijing, China, Sci. Total. Environ.,730,139149,https://doi.org/10.1016/j.scitotenv.2020.139149,2020.

[3] Han,L., Sun,Z.B., He,J., Zhang,B.H., Lv,M.Y., Zhang,X.L., Zheng,C.J.: Estimating the mortality burden attributable to temperature andPM2.5 from the perspective of atmospheric flow. Environ. Res. Lett., 15, 124059, https://doi.org/10.1088/1748-9326/abc8b9,2020.

[4] Han,L., Sun,Z.B., Gong,T.Y., Zhang,X.L., He,J., Xing,Q., Li,Z.M.,Wang,J., Ye,D.X., Miao,S.G.: Assessment of the short-term mortality effect of the national actionplan on air pollution in Beijing, China. Environ. Res. Lett., 15,034052,https://doi.org/10.1088/1748-9326/ab6f13,2020.

[5]Gong,T.Y., Sun,Z.B., Zhang,X.L., Zhang,Y., Wang,S.G., Han,L.,Zhao,D.L., Ding,D.P., Zheng,C.J.:Associations of black carbon and PM2.5 with daily cardiovascular mortalityin Beijing, China.Atmos. Environ., 214, 116876, https://doi.org/10.1016/j.atmosenv.2019.116876,2019.

[6]Hou, Q., An, X.Q., Tao, Y., Sun Z.B.: Assessment of resident's exposure level and health economic costs of PM10 in Beijingfrom 2008 to 2012, Sci. Total. Environ.,563-564,557-565, http://dx.doi.org/10.1016/j.scitotenv.2016.03.215,2016.

42 between health costs and emissions costs based on national or local economic affordability to reduce emissions (Lee et al.,

Comments to the authors: Do you mean "economic costs of controlling emissions"?

**Response to comments:** Yes, our statement is not accurate enough, and relevant contents have been corrected

50 circulation are an important background (Inness et al., 2015). These conditions cause stabilized vertical stratification and low

Comments to the authors: Contributing factor?

**Response to comments:** Yes, our statement is not accurate enough, and relevant contents have been corrected

54 continue to increase (Cai et al.,2017), where the reduction in sea ice can lead to the weakening of the rossby wave activity

Comments to the authors: Run-on sentence, break it up for clarity.

**Response to comments:**

Thanks for the reviewer's suggestion. This sentence has been rewritten as following:

"From a large-scale climate circulation perspective (Markakis et al., 2017; Zou et al., 2017),previous studies have suggested that, if global warming trends continue, the probability of adverse atmospheric pollutant dispersion will continue to increase (Cai et al.,2017).The reduction in sea ice can lead to the weakening of the rossby wave activity south of 40°N, rendering the lower layer colder and a reduced moisture content, a stable atmosphere, weaker wind speeds, and an increased chance of heavy haze pollution(Wang et al.,2015;Chen et al.,2015)."

**60** (Wu et al., 2017). Zhang et al. (2016) use the Kirchhofer technique to classify the circulation patterns, examining the

Comments to the authors: Provide the reference to this technique by Kirchhofer.

**Response to comments:**

The reference has been added.

El-Kadi, A.K.A., Smithson, P.A.: Atmospheric classifications and synoptic climatology, Prog. Phys. Geogr., 16, 432-455, https://doi.org/10.1177/030913339201600403,1992.

61 influence that the monsoon has on the occurrence frequency of different weather patterns and air quality.

Comments to the authors: the air quality associated with those patterns clarify

**Response to comments:**

Thanks for the reviewer's correction. We corrected it as " examining the air quality associated with those patterns clarify"

64 plain wind. Even under conditions associated with weaker synoptic scales, these mesoscale systems largely determine the

Comments to the authors: Surely you mean circulations, not scales?

**Response to comments:**

Thanks, the "scales" was changed to "circulations".

72 have on the concentration of aerosols. A comprehensive analysis of the these two aspects, that is, combining weather

Comments to the authors: The last part of the previous para suggests that these studies also look at the impacts of aerosols on the mesoscale atmospheric structure and circulation.

Remove "the"

**Response to comments:**

Thanks, corrected.

**84** boundary layer is not entirely identical. The above-mentioned weather classification method does not take into account the

Comments to the authors: Unclear. Identical to what? Among different haze episodes? Of Miao et al and Xu et al?

**Response to comments:**

Sorry for the unclear statement. We have rewritten the this part of introduction, and this sentence has been removed in the new version.

100 system(4-D VAR). The model parameters were changed, and the horizontal resolution was enhanced with the use of more

Comments to the authors: Which ones?

**Response to comments:**

"The model parameters were changed, and the horizontal resolution was enhanced with the use of more"

has been modified to

"There are 4 soil moisture layers with the depth of 7cm, 28cm, 100cm and 255cm respectively. The model contains 20 vegetation types, and the land surface parameters change with the change of vegetation types."

101 satellite and ground-based observations(https://apps.ecmwf.int/datasets/).

Comments to the authors: Pertaining to what?

**Response to comments:** This sentence has been deleted.

**124** pollution level (AQI>200). The primary pollutant is PM2.5 in Beijing area. As defined by the AQI, the 24-h average

Comments to the authors: You mean corresponding to these two AQI values? Clarify.

**Response to comments:**

Sorry for the unclear statement. This sentence was corrected as following:

the AQI reaches a moderate pollution level (AQI $\ge$ 150) for more than or equal to 3 days in which at least 1 day reaches the heavy pollution level (AQI $\ge$ 200).

142 layer is more uniform than that of the gaseous pollutants, whereas the particle concentration in the boundary layer is

Comments to the authors: Not clear why gas phase pollutant lifetimes are mentioned here.

**Response to comments:**

We rewrote this sentence as following:

has been modified to

"As the lifetime of a particles is long, that is, several days or weeks, whereas the particle concentration in the boundary layer is generally uniform and significantly different from that in the free atmosphere(Lin et al.,2007;Kang et al.,2019)."

**143** significantly different from that in the free atmosphere. By analyzing the backscattering profile of the atmospheric particles,

Comments to the authors: Reference?

**Response to comments:**

Thanks, reference has been added.

**Reference:**

Lin, C.Y., Wang, Z., Chen, W.N., Chang, S.Y. :Long-range transport of Asian dust and air pollutants to Taiwan:observed evidence and model simulation, Atmos. Chem. Phys., 7, 423–434,

http://doi.org/10.5194/acp-7-423-2007,2007.

Kang, H.Q., Zhu, B., Gao, J.H.,He,Y.,Wang,H.W.,Su,J.F.,Pan,C.,Zhu,T.,Yu,B.:Potential impacts of cold frontal passage on air quality overthe Yangtze River Delta, China. Atmos. Chem. Phys., 19, 3673–3685, http://doi.org/10.5194/acp-19-3673-2019,2019.

**154** with the rotated empirical orthogonal function(REOF) to determine which mode the pollution events belong to according to

Comments to the authors: Cite a reference.

**Response to comments:**

Thanks, references have been added.

Paegle, J. N., Mo, K. C.:Linkages between Summer Rainfall Variability over South America and Sea Surface Temperature Anomalies, Journal of Climate, 15(12):1389-1407, http://doi.org/10.1175/1520-0442(2002)0152.0.CO;2,2002.

Li, J.B., Cook, E. R., D'arrigo, R., Chen, F. H., Gou, X. H.:Moisture variability across China and Mongolia: 1951-2005. Climate Dynamics, 32(7-8):1173-1186, http://doi.org/10.1007/s00382-008-0436-0,2009.

**156** weather. In the empirical orthogonal function (EOF) analysis, the first few main components are the focus of the analysis

Comments to the authors: Rewrite this sentence, not readable.

**Response to comments:**

This sentence has been rewritten as following:"The 925 hPa geopotential height field is affected by both synoptic and local circulations, which can simultaneously reflect the variation characteristics of weather system and boundary layer. Thus, the 925hPa geopotential heights of all pollution events were analyzed in this study by using the 6h ERA-Interim reanalysis data. With 500 samples (4 times each day in 125 days), the rotated empirical orthogonal function (REOF) was used to determine which mode the pollution events belong to according to the characteristic values of the different pollution events. "

**168** In this study, the 925hPa geopotential height was used to classify the pollution weather types into four categories with the

Comments to the authors: Lower panels of the figure appear to be geopotential (gpm). If geopotential height is shown in the figure, it should be in meters.

**Response to comments:**

Thanks, we replot this figure and change the unit to meters.

**175** 43.69, 33.68, 16.51, and 3.92%, respectively (Fig. 2). This indicates that an objective weather classification can effectively

Comments to the authors: The way the figure presents these discrete data as a continuous line is misleading. Present the figure as the data allow. If there are additional data points that went into constructing the interpretation variance plot describe where obtained.

The text jumps around between figures! Better organization is needed to provide the description of the figures in a sequential flow. Also it would be a good idea to present these performance metrics in a table instead of lengthy word descriptions.

**Response to comments:**

The picture has been modified.

Figure 2.The four pollution weather types as a function of their (a) number of samples,(b) proportion with respect to the total number of samples, and (c)interpretation variance.

Thanks for the advice. We are very sorry for the problems caused by our failure to match the pictures with the text.In the previous text, Fig. 2 was used uniformly, without distinguishing between Fig. 2a, Fig. 2b and Fig. 2c, which may make the text lack pertinence. Now we have marked the corresponding picture for each sentence.

**It has been modified to**

"In this study, we observed 125 days of heavy polluted weather. Among these days, type1, type2, type3, and type4 had 67, 27, 21, and 10 days, respectively (Fig.2a), where the four weather types accounted for 53.6, 21.6, 16.8, and 8.0% of the total sampled weather event days, respectively (Fig.2b). The total interpretation variance of the four types for all events was 97.8% while the independent interpretation variance was 43.69, 33.68, 16.51, and 3.92%, respectively (Fig. 2c)."

177 As shown in Fig. 1, the Beijing area is located toward west of the high-pressure system that has its center located in the sea.

Comments to the authors: Briefly explain how the data were sampled to produce this and the subsequent plots. Single-day (which one and how selected) or multi-day averages (over what days)? Which panel of Fig. 1?

**Response to comments:**

"Among 125 days, type1, type2, type3, and type4 had 67, 27, 21, and 10 days, respectively. Multi-day averaged ERA-Interim data was used to produce sea level pressure and geopotential height of925hPa of four types."

The above content has been added.

183 pollutant convergence into the plains along the Taihang Mountains, When type3 appears, the high pressure center was

Comments to the authors: Tense mismatch, rewrite.

**Response to comments:**

Thank you for the reviewer's carefulness. We corrected it as following:

" When type3 appears, the high pressure center is located in the middle of Mongolia, where Beijing is in the front of the weak high pressure system, with a northwest current at 925hPa.However, the wind speed is lower than that affected by strong cold air, because of which it is difficult to penetrate the lower layer of the boundary layer and the wind can only exist in the upper atmosphere of the boundary layer. "

185 at 925hPa(Fig. 1i). However, the wind speed was lower than that affected by strong cold air, because of which it was

Comments to the authors: Wrong label? Fig. 1i seems to be a topo map. The figure caption is missing the description of Fig. 1i.

**Response to comments:**

At the beginning of this paragraph, "As shown in Fig. 1" has been put forward and (Fig. 1i) has been deleted.

188 of Mongolia and southern Hebei province, where there is only a low pressure system with a smaller spatial and temporal

Comments to the authors:Does this refer to Fig 1d? Rewrite this sentence to clearly indicate that western Mongolia is a high pressure region and Southern Hebei province is a low pressure area. As written in the first half of the sentence, they could both be understood to be high pressure regions.

**Response to comments:**

Yes, this refer to Fig 1d.

Following the reviewer's suggestion, this sentence has been rewritten as: When type4 appears, the center of the high pressure system is located further to the northwest of Mongolia, while only a low pressure system is in the southern Hebei province with a smaller spatial and temporal scale

As can be seen from the Figure 1d and 1h, Beijing area is far from the high pressure system, and the low pressure area is not close to Beijing, so there is no obvious weather system affecting Beijing area. Another evidence is that the wind speed at 925hPa is very small, and the 24-hour wind speed in the boundary layer in Figure 4d is also relatively small. Based on the above evidence, we believe that when Type4 appeared, there was no obvious weather system in Beijing, which resulted in a very small pressure gradient and very small wind speed in boundary-layer, which was very conducive to the development of local circulation.

191 which results in a smaller synoptic-scale pressure gradient in Beijing and the surrounding areas(Fig. 1i). Most areas in North

Comments to the authors: Is Fig. 1i the correct figure displaying the synoptic-scale pressure gradient? What I see is topography height. Is something else intended in referring to this figure?

Also the font and figure size for 1i are inconsistent with the rest of the panels, and there is no text in the

figure caption corresponding to 1i.

**Response to comments:**

It should be Fig.1d and has been corrected.

Fig. 1i has been added.

"Most areas in North China do not have strong weather systems and the average wind speed in the boundary layer is smaller, which is favorable to the formation and maintenance of the local circulation considering the topography in the Beijing area (Fig. 1i)."

203 is different, where the inversion height of this type is between 700 and 800hPa.

Comments to the authors: What is that mechanism? It's been described before, but does this figure help illustrate it further?

**Response to comments:**

The description here is not accurate, so we delete the "The mechanism of the thermal structure".

204 As shown in Fig. 4, the basic flow is the southerly wind below 2,000m in type1, where a southwest wind appears from 500–

Comments to the authors: The geographic map is not visible in the background. Is it the same area as in Figure 1?

**Response to comments:**

The mean wind profiles in Fig.4 are observed with a wind profiler radar at the Beijing Observatory. We have added more detailed information for figure 3 to figure 6 in the first paragraph in the section 3.2. The vertical structure of the atmosphere is very important for the formation and evolution of extreme haze events. The vertical thermal and dynamic structures of four weather types are investigated in three-dimensional view. Figure 3 to Figure 6 presented the vertical distribution of temperature, wind and RH, respectively. The temperature in Fig. 3 and vertical speed in Fig.5 are averaged for each weather type by using the 6h ERA-Interim reanalysis data, respectively. The mean wind profiles in Fig.4 are observed with a wind profiler radar at the Beijing Observatory, and the mean relative humidity in Fig.6 is measured with a microwave radiometer at the same site.

In 2.1 Meteorological data, there is description of wind profile data.

" A 12-channel (5water channels and 7oxygen channels) microwave radiometer (Radiometrics, Romeoville, IL, U.S.A.) was used to measure the relative humidity and temperature profile in the atmosphere. The microwave radiometer was installed in the Beijing Observatory ( $39.93^\circ$ N,  $116.28^\circ$ E) and was calibrated every three months. The wind profiles, including the wind speed and direction between 100 and 5,000m, are measured at the same station by a wind profiler. The wind profiler radar provides a set of profile data every 6min at a detection height of ~12–16km."

**This information is also added in the figure captions:**

" Figure 4.Observedmean wind field characteristics of the four pollution weather types in the Beijing area(varying colors, based on the color bar to the right of each panel, represent the wind speed in m s-1; the x-axis is in Beijing time from 00:00 to 23:00; the y-axis is the height in m)."

Comments to the authors: Below 500m? Above that it's southwesterly flow and at 2000m is clearly from the west.

**Response to comments:**

Yes, As shown in Fig. 4, the basic flow is the southerly wind below 2,000m in type1, where a southwest wind appears from 500–2,000m. The south wind is below 500m between04:00 and 20:00, and the easterly wind appears at other times. The south wind speed below500m is 4-6 ms-1 which is higher than the easterly wind(2-4 m s-1)

205 2,000m. The southerly wind is below 500m between 04:00 and 20:00, and the easterly wind appears at other times. The

Comments to the authors: State time standard used (Beijing time).

**Response to comments:**

In 2.1 Meteorological data, " In data analysis, Beijing Local Time (BJT) was used." has been added. Time standard has been added, such as "04:00" has been changed to "04:00 BJT".

206 southerly wind speed at 500m is strong, while the easterly wind is weak. In type2, the basic flow above 1,000m is westerly

Comments to the authors: Better to quantify "strong" and "weak" by providing the ranges of values. **Response to comments:**

Thanks for the reviewer's suggestion. This sentence has been rewritten as :

The south wind speed below500m is 4-6 m s-1 which is higher than the easterly wind(2-4 m s-1).

213 above 500m originates from the northwest from 04:00 to 14:00. At altitudes below 500 m, the wind is southerly and

Comments to the authors: during these hours...

**Response to comments:**

Thanks, corrected.

218 local circulation and basic airflow). Westerly or weak northerly winds above 1,000m in type4 control the atmosphere, where

Comments to the authors: Synoptic?

**Response to comments:**

"basic airflow" has been changed to " synoptic airflow".

224 Figure 5 shows that, below 700hPa, type1, type2, and type4 are ascending movements. The maximum of the synoptic scale

Comments to the authors: Vertical axis label should be changed from "Height" to "Pressure". Change "below" to "above" accordingly in the text below.

**Response to comments:**

Thanks. The picture and text have been modified.

227 sinking movement increases gradually with decreasing height, where the maximum of the sinking movement appears at 900–

Comments to the authors: pressure, not height.

**Response to comments:**

Thanks, corrected.

230 all types, resulting in type3 characterized by the smallest capacity among the four types. Comments to the authors: atmospheric capacity

**Response to comments:**

Thanks, corrected.

231 Based on Fig. 6, the relative humidity profiles for the four weather types have both similarities and differences in their space-

Comments to the authors: Change right-axis label of the figure to say RH. Define RH in the figure caption.

To be consistent with the text the height axis needs to be in meters, not km. Easiest if label is changed to show a 103 scaling,

**Response to comments:**

The figure has been modified.

232 The similarities in the four types are the increased and decreased relative humidity below 1,000m during the night and day, respectively, with a reverse in the relative humidity layer appearing during the day.

Comments to the authors: Does not seem to hold for Type 4 above 2,000 m.

**Response to comments:**

The analysis here is intended to show the inverse humidity at an altitude about 500m for four types. We changed this sentence as:

"The similarities in the four types are the increased and decreased relative humidity below 1,000m during the night and day, respectively, with a reverse in the relative humidity appearing at an altitude about 500m during the day."

234the surface layer decreases daily from 10:00 to 20:00 with an increase in the solar radiation. The thickness of the dry layer in

Comments to the authors: hours Beijing time. Specify this after time references throughout.

**Response to comments:**

Relevant content has been revised.

236 The top of the dry layer is the reverse of the relative humidity layer. Above 1,000m, the relative humidity of the other three types, except type2, decreases significantly during the day

Comments to the authors: Explain what is happening in Type 4 above 2000 m.

**Response to comments:**

As mentioned in section 3.1, in type4, Beijing is located between a high pressure and a low pressure and in the front of the weak frontal zone. Stratus cloud with high stability is located in the front of the weak frontal zone above 2000m, so relative humidity above 2,000m is high.

246 profile in the pollution boundary layer formed under the condition of wave-current interaction in the atmosphere (Fig. 6).

Comments to the authors: Not clear what this reference to the figure indicates.

**Response to comments:**

The Fig. 6 indicated the fluctuation feature of the basic flow and was moved after this sentence. The last sentence is a summary analysis.

247 Type2 has strong westerly characteristics (Fig. 4), which reflects more baroclinic characteristics in the atmospheric vertical

Comments to the authors: At what height(s)?

**Response to comments:**

We added the height in the revised version as following:

Type2 has strong westerly characteristics below 3,000 m (Fig. 4), which reflects more baroclinic characteristics in the atmospheric vertical structure for the westerlies.

259 type4: local accumulation(Fig. 4). When type 1 appears, the Beijing area is located at the rear of the high-pressure system,

Comments to the authors: Should this reference be to Fig. 7? Fig. 4 as mentioned before contains no geographic map, making it difficult to understand references to geographic features in the following text.

**Response to comments:**

Yes, this reference has modified to Fig. 7. Figure 4 is a mean diurnal wind profile observed at the Beijing Observatory, so there is no need to show the terrain, which has already been explained in above answers.

260 consistent with southerly winds throughout the atmosphere, and multilayer inversion occurs in the boundary layer. Under the

Comments to the authors: That is not obvious in panel 1 of Fig. 4!

**Response to comments:**

Maybe we didn't explain it clearly. It can be seen from Type1 in Figure 1 that all the winds in Beijing and surrounding areas have the component of southerly wind. The panel 1 in Fig.4 indicated that the southerly wind is prevailing below about 2000m. To showed clearly, we added Fig.1e and Fig.4 in this sentence.

262 the Hebei region have evident regional transport features. When type2 appears, the Beijing area is located at the bottom of

Comments to the authors: Evident from which figure? Fig. 1?

**Response to comments:**

"The air pollutants in the Hebei region have evident regional transport features (Fig. 1)."

263the high-pressure system, where the air above 850hPa is a westerly wind, with easterly winds below 850hPa. Under the

Comments to the authors: As shown in Fig 4, where the vertical axis is in meters, not hPa? In fact, I can find no figure that shows this.

**Response to comments:**

Because the vertical axes are different for the wind profile and temperature profile, we chose the pressure axis, which is widely used in meteorology, to make conceptual model. In Beijing area, 850hPa is generally located at a height of about 1500m, which is explained in this revised version.